# Unrealized Expectations: Comparing AI Methods vs Classical Algorithms for Maximum Independent Set

**Yikai Wu**[*]                                                          *yikai.wu@cs.princeton.edu*
*Department of Computer Science & Princeton Language and Intelligence (PLI)*
*Princeton University*

**Haoyu Zhao**[*]                                                        *haoyu@cs.princeton.edu*
*Department of Computer Science & Princeton Language and Intelligence (PLI)*
*Princeton University*

**Sanjeev Arora**                                                        *arora@cs.princeton.edu*
*Department of Computer Science & Princeton Language and Intelligence (PLI)*
*Princeton University*

**Reviewed on OpenReview:** *https://openreview.net/forum?id=ksGoCT5zW6*

## Abstract

AI methods, such as generative models and reinforcement learning, have recently been applied to combinatorial optimization (CO) problems, especially NP-hard ones. This paper compares such GPU-based methods with classical CPU-based methods on *Maximum Independent Set* (MIS). Strikingly, even on in-distribution random graphs, leading AI-inspired methods are consistently outperformed by state-of-art classical solver `KaMIS` running on a single CPU, and some AI-inspired methods frequently fail to surpass even the simplest *degree-based greedy* heuristic. Even with post-processing techniques like local search, AI-inspired methods still perform worse than CPU-based solvers. To better understand the source of these failures, we introduce a novel analysis, *serialization*, which reveals that *non-backtracking* AI-inspired methods, e.g. `LTFT` (which is based on GFlowNets), end up reasoning similarly to the simplest degree-based greedy, and thus worse than `KaMIS`. More generally, our findings suggest a need for a rethinking of current approaches in AI for CO, advocating for more rigorous benchmarking and the principled integration of classical heuristics. Additionally, we also find that CPU-based algorithm `KaMIS` have strong performance on sparse random graphs, which appears to show that the shattering threshold conjecture for large independent sets proposed by Coja-Oghlan & Efthymiou (2015) does not apply for real-life sizes (such as $10^6$ nodes).

## 1 Introduction

Combinatorial optimization (CO) lies at the core of numerous scientific and engineering studies, encompassing applications in network design, resource allocation, healthcare, and supply chain (Du & Pardalos, 1998; Hoffman, 2000; Zhong & Tang, 2021). Combinatorial optimization usually involves selecting an optimal solution from a discrete but often exponentially large set of candidates. Many are NP-hard (meaning that if $P \neq NP$ then there is no polynomial-time algorithm for solving them in the general cases (Papadimitriou & Steiglitz, 1998)). This makes it challenging to design algorithms with provable guarantees, but in practice solvers can find reasonable quality solutions (e.g., `Gurobi` (Gurobi Optimization, LLC, 2024)).

Recent advances in artificial intelligence (AI) and GPU computing have motivated use of AI-inspired approaches, e.g., Graph Neural Netwoks (GNNs) and reinforcement learning, to learn problem-specific strategies for NP-hard optimization problems such as MIS (Li et al., 2018; Ahn et al., 2020) and TSP (Kool

---

[*]Equal contribution. Correspondence to: Yikai Wu ( `yikai.wu@cs.princeton.edu`).

et al., 2018; Zhang et al., 2021). AI models can also be trained to predict search directions or refine heuristic rules (Li et al., 2018; d O Costa et al., 2020). These algorithms utilize advanced GPUs and often require shorter inference compared to CPU-based algorithms. Additionally, AI-inspired methods avoid the need to design heuristics for specific problems, allowing generalization to new instances and problems (Bengio et al., 2021b; Cappart et al., 2023).

Despite the claimed benefits of AI-inspired methods, a few years ago Angelini & Ricci-Tersenghi (2023) showed that one specific GNN-based MIS algorithm (Schuetz et al., 2022) failed to surpass greedy algorithms. While their work focused specifically on evaluating that particular solver, it raised critical questions about the baseline performance of AI-based methods. Böther et al. (2022) showed that AI-inspired approaches fail to provide superior search directions compared to traditional heuristics in tree search algorithms for MIS. Gamarnik (2023) suggests that GNN has theoretical limits which may become obstacles for GNN-based MIS algorithms.

However, in recent years many new AI-inspired CO algorithms, utilizing a variety of techniques, such as diffusion models (Sun & Yang, 2023; Sanokowski et al., 2024), GPU-accelerated sampling (Sun et al., 2023), and direct optimization (Alkhouri et al., 2025) have been developed and claimed to significant improve over the previous ones. Furthermore, GFlowNets Bengio et al. (2021a), which have been proposed as general-purpose tools for tasks like scientific discovery and reinforcement learning Bengio et al. (2023), has also been used to solve CO problems (Zhang et al., 2023). Since combinatorial optimization is an arena where humans have hand-designed algorithms for many decades, the following question is of great scientific interest:

> *Do AI-inspired algorithms perform better than classical heuristics for combinatorial optimization?*

## 1.1 Our contributions

We explore the question in the context of `Maximum Independent Set` (MIS) problem: given a graph, aiming to find the largest subset of nodes with no edges present between any node pair. The simplicity of the problem attracted design of many heuristics to tackle the problem (Andrade et al., 2012; Lamm et al., 2015). In recent works, MIS is also a main target in efforts that design AI-inspired approaches such as non-convex optimization (Schuetz et al., 2022; Alkhouri et al., 2025), reinforcement learning (Ahn et al., 2020; Zhang et al., 2023), and diffusion models (Sun & Yang, 2023; Sanokowski et al., 2024).

For classical heuristics, we test degree-based greedy (`Deg-Greedy`, pick a node with smallest degree at each step), and the state-of-the-art MIS solver `KaMIS` (Lamm et al., 2017; Dahlum et al., 2016). For AI-inspired algorithms, we test the newest algorithms from each "category" according to the techniques they use, including sampling algorithm `iSCO` (Sun et al., 2023), non-convex optimization algorithm `PCQO` (Alkhouri et al., 2025), reinforcement-learning related algorithms `LwD` (Ahn et al., 2020) and `LTFT` (Zhang et al., 2023) (based on GFlowNets (Bengio et al., 2021a)) and diffusion models `DIFUSCO` (Sun & Yang, 2023) and `DiffUCO` (Sanokowski et al., 2024).

Testing on different graph types with different sizes and densities leads to the following empirical finding (Section 3).

**Finding 1**: *Current AI-inspired algorithms for MIS still don't outperform the best heuristic `KaMIS`, which runs on a single thread in a CPU, while AI-inspired methods often require significant computational resources.*

**Finding 2**: *As the graph becomes larger or denser, `KaMIS` exhibits a notable superiority to AI-inspired algorithms.*

**Finding 3**: *The simplest degree-based greedy algorithm (`Deg-Greedy`) serves as a very strong baseline. Some AI-inspired algorithms perform similarly to or worse than `Deg-Greedy`, especially for larger and denser graphs.*

Section 4 presents ablation studies to understand why some AI-inspired methods fail to improve over the simplest `Deg-Greedy` method. We introduce a new mode of analysis, *serialization*, that transforms the solution of any algorithm into a sequential list of choices leading to the final independent set. We compare sequential order with the one produced by `Deg-Greedy` (Sections 4.1 and 4.2). We find that the reinforcement

learning related algorithm `LTFT`, based on GFlowNets, behaves similarly to `Deg-Greedy`. We also found several qualitative characteristics that appear to distinguish algorithms that perform significantly better than `Deg-Greedy` from those that perform similarly or even worse than `Deg-Greedy`. We also explore whether AI-inspired method can improve their solution quality via a post-processing step using local search (Section 4.3), but find that they still fail to outperform `KaMIS` after some improvements. In Section 4.4, we discuss an additional result that may be of interest for MIS experts: on random graphs, `KaMIS` has a level of performance on MIS showing that the shattering threshold conjecture for large independent sets proposed by Coja-Oghlan & Efthymiou (2015) does not apply for real-life sizes (such as $10^6$ nodes).

Details on the code, external codebases, and data generation procedures are provided in Appendix D. Our implementation is publicly available at `https://github.com/yikai-wu/MIS-UnExp`.

## 2 Benchmarking Classical and AI-inspired Methods for Maximum Independent Set

We focus the experiment setup for benchmarking different algorithms for Maximum Independent Set problems (MIS).

### 2.1 Maximum Independent Set (MIS) problem

Given an undirected graph $\mathcal{G}(\mathcal{V}, \mathcal{E})$ where $\mathcal{V}$ is the set of nodes and $\mathcal{E}$ is the set of edges, an *independent set* is a subset of vertices $\mathcal{I} \subseteq \mathcal{V}$ such that no two nodes in $\mathcal{I}$ are adjacent, i.e., $(u, v) \notin \mathcal{E}$ for all $u, v \in \mathcal{I}$. The goal in MIS is to find the largest possible independent set, $\mathcal{I}^*$.

### 2.2 MIS algorithms

We classify the algorithms we test as: (1) *classical heuristics*, which includes `Deg-Greedy` and `KaMIS` (`OnlineMIS` (Dahlum et al., 2016) and `ReduMIS` (Lamm et al., 2017)); (2) GPU-accelerated non-learning algorithms, which includes `iSCO` (Sun et al., 2023) and `PCQO` (Alkhouri et al., 2022); and (3) *learning-based* algorithms, which includes `LwD` (Ahn et al., 2020), `LTFT` (Zhang et al., 2023), `DIFUSCO` (Sun & Yang, 2023) and `DiffUCO` (Sanokowski et al., 2024). We provide brief introductions of them below, please refer to Appendices B.4 to B.6 for details.

`Deg-Greedy` (Degree-based greedy) Simplest heuristic: always picks a node with the smallest degree in the current graph, add to the current independent set, and delete that node and all of its neighbors from the graph. Most papers on AI-inspired methods do not compare with this baseline.

`OnlineMIS and ReduMIS` are two variants of the MIS solver `KaMIS`, mainly consists of three alternating steps: greedy, local search, and graph reductions. `OnlineMIS` (Dahlum et al., 2016) only applies a simple reduction after local search, while `ReduMIS` Lamm et al. (2017) applies many graph reduction techniques.

`iSCO` (Sun et al., 2023) is a GPU-accelerated sampling-based method, incorporating gradient-based discrete MCMC and simulated annealing. The MCMC is designed based on the Metropolis-Hasting algorithm, which if given enough time (exponential), can get the optimal solution.

`PCQO` (Alkhouri et al., 2025) directly optimizes the quadratic loss function of the MIS using gradient descent. It is sensitive to optimization hyperparameters, so hyperparameter search is required for achieving good results. Extensive hyperparameter search may give better results than in our experiments.

`LwD` (Ahn et al., 2020) is a reinforcement learning based algorithm which models the problem as a Markov Decision Process (MDP) and requires a dataset (without solutions) to train the policy. In each step, several nodes are added to the independent set and are never deleted. We call it a *non-backtracking* algorithm. Böther et al. (2022) also benchmarked this algorithm. Ahn et al. (2020) reported that it outperforms `KaMIS` on very large but very sparse random graphs, which we do not include in our benchmark, while Böther et al. (2022) found that their performance are similar on these graphs.

`LTFT` (Zhang et al., 2023) is also a *non-backtracking* MDP-based algorithm similar to `LwD`, but it only selects one node at a time, which is decided by GFlowNets (Bengio et al., 2021a). Thus, it has a very similar procedure to `Deg-Greedy`. The algorithm requires a dataset (without solutions) to train the neural network.

`DIFUSCO` (Sun & Yang, 2023) is an end-to-end *one-shot* MIS solver using diffusion models and requires a supervised training dataset with solutions.

`DiffUCO` (Sanokowski et al., 2024) also uses diffusion model but with unsupervised learning and annealing techniques. It requires a training dataset without solutions.

**Non-backtracking vs one-shot** Among algorithms that build up the set step by step, `Deg-Greedy`, `LTFT` and `LwD` are *non-backtracking*, meaning once a node is added to the set it is never dropped from it. `OnlineMIS` and `ReduMIS` are *backtracking* algorithms, since as part of local search they might decide that a previously picked should be dropped from the set to allow further additions. AI-inspired methos `PCQO`, `DIFUSCO`, and `DiffUCO` are *one-shot* algorithms, since they work like end-to-end MIS solvers and directly return the full set.

## 2.3 Graph types

**Erdős-Renyí (ER) graphs** (Erdös & Rényi, 1959) are random graphs where edges are connected uniformly at random (with a given probability or a fixed number of edges). We vary 2 parameters for ER graphs, number of nodes $n$ and average degree $d$, by fixing the number of edges at $\frac{nd}{2}$. Previous benchmark (Böther et al., 2022) and algorithms (Ahn et al., 2020; Sun & Yang, 2023; Zhang et al., 2023; Alkhouri et al., 2025) used it as test graphs for MIS, though without varying parameters as we did.

**Barabási–Albert (BA) graphs** (Albert & Barabási, 2002) are random graphs generated by a probabilistic growth process, mimicking real-world networks such as Internet, citation networks, and social networks (Albert & Barabási, 2002; Radicchi et al., 2011). For BA graphs, we vary 2 parameters: number of nodes $n$ and parameter $m$ (not number of edges). The average degree of BA graphs can be approximated as $2m$.

**RB graphs** RB graphs are derived from Model RB (Xu & Li, 2000), a random constraint satisfaction problem (CSP) model. RB graphs are considered difficult instances for MIS due to their structured randomness and high solution hardness. We use two datasets from Zhang et al. (2023) (also used in Sanokowski et al. (2024)), `RB-small` (200–300 nodes) and `RB-large` (800–1200 nodes), to benchmark learning-based solvers.

**Real-world graphs** We pick REDDIT-MULTI-5K and COLLAB (Yanardag & Vishwanathan, 2015) from TUDataset website (Morris et al., 2020), since they have enough graphs for training and graph sizes not too small. REDDIT-MULTI-5K has 508.52 average nodes and 594.87 average edges. They are mostly very sparse graphs. COLLAB has 74.49 average nodes and 2457.78 average edges. They are mostly small but dense graphs.

## 2.4 More experiment details and computational resources

For synthetic graphs, we test 8 instances for each parameter $(n, d)$ or $(n, m)$; for real-world datasets, we test 100 graphs. Learning-based methods are trained on 4000 graphs generated with the same parameter setting (for random graphs) or drawn from the same real-world dataset. Unless otherwise specified, default hyperparameters are used (details in Appendix B).

Our benchmark evaluates solution quality under fixed resource budgets. The computational limits for each method are summarized in Table 1, including hardware, time budget, memory allocation, and the largest graph sizes successfully handled. CPU-based methods (`Deg-Greedy`, `OnlineMIS`, `ReduMIS`) run on a single CPU thread with a 24-hour limit and up to 64GB RAM; in practice, `ReduMIS` often terminates much earlier on smaller graphs but can scale to $n \approx 10^6$. GPU-based and learning-based methods are evaluated using A100 GPUs with a 96-hour limit (including training), and their scalability is constrained by GPU memory and training completion within this budget (see Appendices B.4 to B.6). We use best-of-20 sampling for all learning-based algorithms and for `Deg-Greedy`.

Table 1: **Summary of computational resource limits used in the benchmark.** We report the hardware configuration, time limits, memory allocations, and the largest graph size successfully trained or evaluated under the specified resource constraints. Memory refers to RAM for CPU-based methods and GPU memory for GPU-based methods. The values represent the *resource limits allocated in our benchmark setup*, not the actual runtime or peak memory consumption required by the algorithms. Among the evaluated methods, only `DiffUCO` supports multi-GPU training.

| Methods | Hardware | Time Limit | Memory | Graph Successfully Trained or Evaluated |
|---|---|---|---|---|
| `Deg-Greedy`, `OnlineMIS`, `ReduMIS` | 1 CPU thread | 24 hrs | 64GB | $\geq 10^6$ |
| `iSCO` | $1\times$ A100 | 96 hrs | 80GB | $\leq 10000$ (fails on dense graphs) |
| `PCQO` | $1\times$ A100 | 96 hrs | 80GB | $\leq 10000$ |
| `LwD`, `LTFT`, `DIFUSCO` | $1\times$ A100 | 96 hrs (training) | 80GB | $\leq 3000$ |
| `DiffUCO` | A100 GPUs | 96 hrs (training) | 320GB | $\leq 1000$ |

## 3 Performance Gap: Classical Methods Outperform AI-inspired Methods

In this section, we present our main experiment results. The performance of different algorithms on Erdős-Renyí (ER) graphs, Barabási–Albert (BA) graphs, RB graphs, and real-world graphs are shown in Tables 2 to 5 respectively.

**AI-inspired algorithms don't outperform `ReduMIS`.** Our first main finding is that, current AI-inspired algorithms do not outperform the best classical heuristics `ReduMIS` in terms of performance. As shown in Tables 2 to 5, `ReduMIS` consistently achieves superior results compared to all other methods, with the exception of `iSCO` sometimes perform similarly.

Although learning-based algorithms are claimed to be more efficient, they require significant training time and GPU memory, which is over our resource constraint for graphs with more than 3000 nodes, while `ReduMIS` can handle graphs with up to $1 \times 10^6$ nodes (See Table 9). Although `iSCO` performs close to `ReduMIS`, it requires significant GPU memory and we are unable to get results for dense graphs with 10000 nodes. Its performance also become worse than `ReduMIS` for larger graphs.

We also note that `LwD` performs the best among learning-based algorithms, despite it being the oldest learning-based algorithm we tested. In summary, our experiment results show that current AI-inspired algorithms still don't outperform the best classical heuristics for the MIS problem.

**The performance gap between `ReduMIS` and AI-based methods widens with larger and denser graphs.** While `ReduMIS` consistently outperforms AI-based methods in most cases, the performance gap is small on small or sparse graphs. On ER graphs when $n = 100, 300$ and average degree $d = 10$, `LwD` has results within 1% gap from `ReduMIS`. However, as the graph becomes larger or denser, the performance gap between `ReduMIS` and AI-based algorithms enlarges. On ER graphs, when $n = 1000$, there is a clear performance gap between AI-based algorithms and `ReduMIS`, with the only exception of the sampling based `iSCO`. When $d = 10$, `DiffUCO` and `LwD` still reaches 98% of `ReduMIS`'s performance. For denser graphs($d = 100$), `DiffUCO` and `LwD` only reaches 96% of `ReduMIS`'s performance. This gap widens further for $n = 3000$, where AI-based algorithms perform significantly worse than `ReduMIS` and sometimes fail to outperform simple heuristic `Deg-Greedy`. Classical solvers have no difficulty handling graphs with a million edges, but learning-based implementations struggle to scale up to that size.

**`Deg-Greedy` serves as a strong baseline.** Another key finding is that the simplest degree-based greedy (`Deg-Greedy`) serves as a remarkably strong baseline. As shown in Table 2, leveraging neural networks for

Table 2: **Performance of different algorithms on Erdős–Rényi (ER) graphs.** We report the average independent set size among 8 graphs generated by the graph parameters $n, d$. '–' denotes the algorithm fails to return a solution within 96 hours, or the graph cannot be fitted into the GPU resources: a single 80GB A100 GPU for `iSCO`, `LTFT` and `DIFUSCO`, and four 80GB A100 GPUs for `DiffUCO`. Best-of-20 sampling for `Deg-Greedy` and all learning-based algorithms. The numbers within $\pm1\%$ of the best in each row are highlighted. $*$ denotes training terminated without reaching the target steps and test using the latest checkpoint. † denote testing with out-of-distribution trained models. Details in Appendix B.

| | | CPU-based | | | GPU-acc | | Learning-based | | | |
|---|---|---|---|---|---|---|---|---|---|---|
| $n$ | $d$ | Deg-Greedy | OnlineMIS | ReduMIS | iSCO | PCQO | LwD | LTFT | DIFUSCO | DiffUCO |
| 100 | 10 | 29.25 | **30.50** | **30.50** | **30.62** | **30.63** | **30.38** | 28.62 | 30.25 | 30.02 |
| | 30 | 13.63 | 14.00 | **14.75** | 14.50 | 14.50 | 14.38 | 13.12 | 13.88 | 13.92 |
| 300 | 10 | 77.50 | **93.88** | **94.38** | **94.75** | **94.63** | **94.25** | 88.62 | 93.50 | 91.84 |
| | 30 | 44.50 | 47.88 | **47.88** | **47.62** | **47.63** | 46.88 | 43.25 | 43.88 | 45.04 |
| | 100 | 16.13 | 18.00 | **18.38** | 18.00 | 18.00 | 17.00 | 16.25 | 16.62 | 16.93 |
| 1000 | 10 | 303.25 | **314.75** | **316.13** | **315.62** | 310.00 | 311.25 | 297.00 | 303.88 | 311.67 |
| | 30 | 151.00 | 158.88 | **163.75** | **163.50** | 158.63 | 158.38 | 150.00 | 143.75 | 152.55† |
| | 100 | 60.63 | 64.75 | **66.63** | **66.50** | 60.13 | 63.88 | 60.88 | 55.38 | 63.55† |
| | 300 | 22.25 | 25.00 | **25.75** | 24.62 | 23.25 | 19.12* | 22.62 | 20.88 | 18.36† |
| 3000 | 10 | 907.13 | **947.25** | **954.25** | **950.88** | 923.25 | 934.12 | 888.25 | 902.00 | 935.28† |
| | 30 | 451.88 | 480.88 | **493.13** | **491.62** | 464.25 | 473.25 | 449.00 | 413.38 | 459.57† |
| | 100 | 183.63 | 194.38 | **201.50** | **200.38** | 185.63 | 190.75* | 184.00 | 171.38 | 194.07† |
| | 300 | 73.50 | 77.63 | **80.75** | 78.88 | 69.25 | – | 73.88 | – | – |
| | 1000 | 23.38 | 26.00 | **26.25** | – | 23.00 | – | 23.62 | – | – |
| 10000 | 10 | 2999.88 | **3161.88** | **3173.62** | **3149.92** | 2569.75 | – | – | – | – |
| | 30 | 1498.00 | 1607.50 | **1639.88** | **1625.47** | 1331.63 | – | – | – | – |
| | 100 | 613.75 | 650.00 | **670.88** | – | 543.13 | – | – | – | – |
| | 300 | 249.00 | 258.38 | **272.25** | – | 218.63 | – | – | – | – |
| | 1000 | 87.13 | 91.50 | **94.25** | – | 77.50 | – | – | – | – |
| | 3000 | 29.63 | 33.00 | **33.25** | – | 26.63 | – | – | – | – |

node selection, `LTFT` often perform comparably to `Deg-Greedy`, particularly on larger or denser graphs. For example on ER graphs when $n = 1000$ or $3000$, `LTFT` gives performance within 2% of `Deg-Greedy` (Table 2). Additionally, `DIFUSCO` and `PCQO` fail to outperform `Deg-Greedy` on larger or denser graphs, such as $n = 1000$ with $d \geq 100$ and $n = 3000$ for `DIFUSCO`, $n = 3000$ with $d \geq 300$and $n = 10000$ for `PCQO`.

**Similar trends on other types of graphs** We also tested on Barabási–Albert (BA) graphs (Table 3), RB graphs (Table 4), and real-world graphs (Table 5). For BA graphs and RB graphs, `ReduMIS` also outperform all other methods across various setting. For real-world graphs, most methods including `ReduMIS` perform close to the best, likely because these datasets are either small or sparse and thus easy for MIS problem. On BA and RB graphs, we also observe that the performance gap between `ReduMIS` and AI-inspired methods increase as the graphs become larger and denser.

`Deg-Greedy` is also a strong baseline for all these graph types. Notably, for large RB graphs, all learning-based methods perform worse than `Deg-Greedy`. Since RB graphs are considered difficult cases for MIS (Zhang et al., 2023), these results may suggest learning-based methods are even weaker for graph types with higher "intrinsic hardness" than ER graphs.

**Performance on Real-World Graphs** On real-world graphs, we observe roughly equivalent performance across most methods, including the simplest degree-based greedy heuristic `Deg-Greedy`. This suggests that available datasets suitable for training are algorithmically "easy," making it difficult to differentiate between sophisticated solvers and simple heuristics. Consequently, the primary disadvantage of AI-based methods here is computational efficiency: while `Deg-Greedy` achieves optimal or near-optimal results on a single CPU thread, AI methods require significant GPU resources and orders of magnitude longer time for training

Table 3: **Performance of different algorithms on Barabási–Albert (BA) graphs.** Table 2 while changing ER graphs to BA graphs with graph parameter $n, m$. We report the average independent set size among 8 graphs generated by the graph parameters $n, m$. The numbers within $\pm 0.5\%$ of the best in each row are highlighted. See caption of Table 2 for other details and definitions of $*$ and $\dagger$.

| $n$ | $m$ | Heuristics | | | GPU-acc | | Learning-based | | | |
|---|---|---|---|---|---|---|---|---|---|---|
| | | Deg-Greedy | OnlineMIS | ReduMIS | iSCO | PCQO | LwD | LTFT | DIFUSCO | DiffUCO |
| 100 | 5 | **39.50** | **39.50** | **39.50** | **39.50** | **39.50** | **39.50** | 38.12 | **39.38** | 39.25 |
| | 15 | 21.00 | **21.63** | **21.63** | **21.62** | **21.63** | **21.62** | 20.62 | 21.25 | 21.25 |
| 300 | 5 | 121.88 | **123.13** | **123.13** | **123.12** | **123.00** | **123.12** | 115.62 | **122.88** | **123.00** |
| | 15 | 69.38 | **71.38** | **71.38** | **71.38** | **71.25** | 70.75 | 64.75 | 69.50 | 69.25 |
| | 50 | 38.00 | **49.88** | **50.00** | **50.00** | **50.00** | **50.00** | 41.75 | **50.00** | **50.00** |
| 1000 | 5 | 412.63 | **417.13** | **417.13** | **417.12** | 415.75 | **416.00** | 385.75 | **417.12** | **416.00** |
| | 15 | 232.75 | 245.00 | **246.38** | **246.25** | 242.00 | 243.12 | 224.25 | 236.38 | 240.75 |
| | 50 | 107.50 | 115.75 | **116.88** | **116.75** | 113.50 | 113.00* | 106.38 | 105.25 | 56.38† |
| | 150 | 81.50 | 150.00 | **150.00** | **150.00** | **150.00** | **150.00*** | 82.25 | – | 135.00† |
| 3000 | 5 | 1236.63 | **1257.00** | **1257.13** | **1255.62** | 1243.00 | 1248.12 | 1177.25 | **1254.38** | **1252.63†** |
| | 15 | 709.50 | 749.63 | **754.50** | **752.00** | 722.25 | 727.75* | 661.88 | 726.38 | 737.38† |
| | 50 | 335.38 | 362.63 | **369.75** | **368.25** | 357.13 | 336.88* | 323.88 | – | 142.25† |
| | 150 | 147.25 | 160.25 | **165.75** | 164.00 | 152.75 | – | 144.75 | – | – |
| | 500 | 141.00 | **500.00** | **500.00** | – | 491.00 | – | 223.88 | – | – |
| 10000 | 5 | 4128.75 | **4206.00** | **4205.38** | **4190.47** | 3640.75 | – | – | – | – |
| | 15 | 2377.88 | **2525.00** | **2534.00** | 2517.07 | 1946.38 | – | – | – | – |
| | 50 | 1122.13 | 1228.88 | **1251.13** | 1241.98 | 938.38 | – | – | – | – |
| | 150 | 511.25 | 555.75 | **580.13** | 573.37 | 433.88 | – | – | – | – |
| | 500 | 192.50 | 209.13 | **217.50** | – | 150.63 | – | – | – | – |
| | 1500 | 67.25 | **1500.00** | **1500.00** | – | – | – | – | – | – |

Table 4: **(Comparison of the performance of different algorithms on RB graphs)** Table 2 on RB graphs (Xu & Li, 2000). Datasets are from Zhang et al. (2023). The numbers within $\pm 1\%$ of the best are highlighted.

| Dataset | no. of nodes | Heuristics | | | GPU-acc | | Learning-based | | | |
|---|---|---|---|---|---|---|---|---|---|---|
| | | Deg-Greedy | OnlineMIS | ReduMIS | iSCO | PCQO | LwD | LTFT | DIFUSCO | DiffUCO |
| RB-small | 200-300 | 19.11 | **20.11** | **20.14** | **20.05** | **20.08** | 16.44 | 18.63 | 18.23 | 19.26 |
| RB-large | 800-1200 | 39.16 | **42.66** | **42.95** | 41.53 | 39.62 | 32.55 | 38.32 | 36.22 | 38.91 |

Table 5: **(Comparison of the performance of different algorithms on real-world graphs)** Table 2 on real-world datasets. In general, graphs in REDDIT-MULTI-5K are very sparse, while COLLAB are dense but small. The numbers within $\pm 1\%$ of the best are highlighted.

| Dataset | Heuristics | | | GPU-acc | | Learning-based | | | |
|---|---|---|---|---|---|---|---|---|---|
| | Deg-Greedy | OnlineMIS | ReduMIS | iSCO | PCQO | LwD | LTFT | DIFUSCO | DiffUCO |
| REDDIT-MULTI-5K | **350.73** | **350.73** | **350.66** | **350.73** | 344.47 | **350.73** | 343.35 | **350.72** | **350.69** |
| COLLAB | **8.68** | **8.70** | **8.70** | **8.70** | 8.57 | **8.69** | **8.70** | **8.75** | **8.70** |

and inference. Given the scarcity of harder real-world datasets that are suitable for training (sufficient quantity/size), we followed the community standard of using random graph (ER and BA) for our main evaluation.

## 4 Deconstructing the Performance Gap: Algorithmic Analysis

Our results show that the state-of-the-art AI-inspired algorithms for MIS still do not outperform the best heuristic `ReduMIS`. The surprising finding was that they also often do not outperform the simplest classical

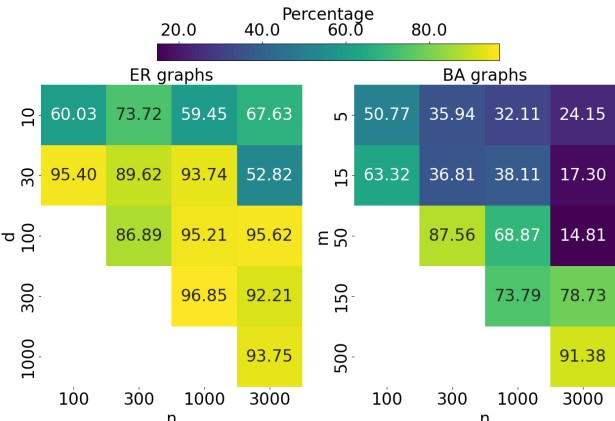

Figure 1: **Percentage of rounds when LTFT selects the node with smallest possible degree, i.e., behaves similarly to degree-based greedy.** On ER graphs when the graph is dense (closer to the bottom left corner), the percentage of rounds LTFT selects the node with smallest possible degree is higher, i.e., behaves more similarly to degree-based greedy. On BA graphs, the percentage to choose the smallest possible node is generally lower, but on denser BA graphs LTFT also behaves more like degree-based greedy.

heuristic, Deg-Greedy, especially on large and dense graphs. In this section, we delve deeper into this comparison (Sections 4.1 and 4.2). Furthermore, we explore the impact of augmenting various algorithms with a local search as a post hoc step to enhance solution quality (Section 4.3).

## 4.1 Comparison between Deg-Greedy and LTFT

Deg-Greedy sequentially picks nodes for the independent set. At each step, it picks the node with the smallest degree in the residual graph (where the nodes in the independent set and their neighbors are removed). It does not reverse any decisions (ie once picked, the node stays in the independent set). As in Section 2.2 we call it a *non-backtracking* algorithm. LTFT is also a non-backtracking algorithm and it often perform similarly to Deg-Greedy in Tables 2 and 3. It uses a trained policy network GFlowNets (Bengio et al., 2021a) to pick a node for the independent set at each step. Thus, we can naturally compare LTFT with Deg-Greedy by investigating how close this trained policy compares to the naive policy in Deg-Greedy.

The results are shown in Figure 1. Overall, we observe that LTFT frequently selects nodes with very small degrees. On ER graphs where the average degree is at least 30, LTFT picks the node with the smallest degree over 85% of rounds except for sparse graphs ($d = 10$ and $(n, d) = (3000, 30)$). On BA graphs, while the percentage is lower, it still exceeds 75% on large and sufficiently dense graphs. Moreover, in cases where LTFT selects the smallest degree nodes less frequently, its performance is worse than Deg-Greedy.

To conclude, our results suggest that despite using neural net to learn the policy, LTFT is closely aligned with Deg-Greedy: prioritizing nodes with small degrees at each step. The high consistency between the node selection strategies of LTFT and Deg-Greedy can explain their similar performance.

## 4.2 Serialization: allows comparing to Deg-Greedy

In the previous part, we demonstrate that LTFT employs a heuristic similar to Deg-Greedy, selecting the node with the smallest degree in the remaining graph in each round. However, other algorithms, such as ReduMIS, PCQO, and DIFUSCO, which does not select one node at a step without backtracking, cannot be analyzed based on the sequence of nodes picked. Thus, we introduce a method called *degree-based solution serialization* to analyze their behavior and compare them with Deg-Greedy.

Given a graph $\mathcal{G}(\mathcal{V}, \mathcal{E})$ and an independent set solution $\mathcal{I}$ (which is an independent set), process proceeds as follows: (1) Repeatedly remove the node in $\mathcal{I}$ with the smallest degree. (2) After removing a node from $\mathcal{I}$,

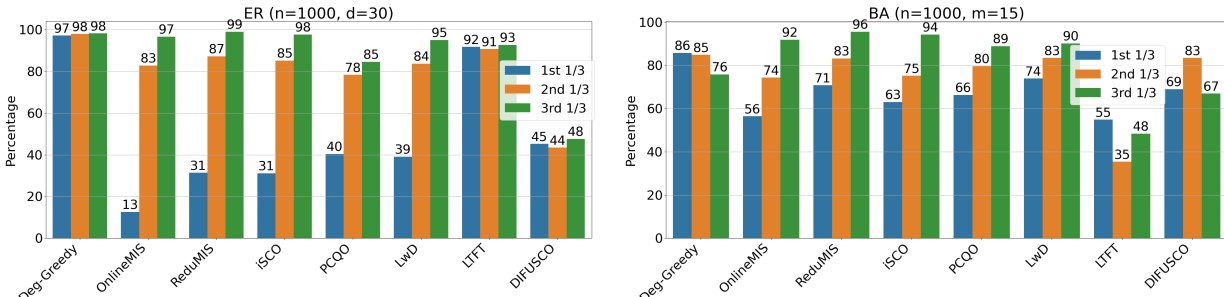

Figure 2: **The percentage to choose the smallest possible degree node on different part of the (degree-based) serialization.** We find that the best algorithms (`OnlineMIS`, `ReduMIS`, `iSCO`) and the best performing learning-based algorithm `LwD` share a similar characteristic pattern, that they have high consistency with degree-based greedy on second and third part of the serialization, while on the first part there is a low consistency. On the other hand, `PCQO` and `DIFUSCO` has low consistency with degree-based greedy in general.

also remove it and its neighbors in $\mathcal{G}$. (3) Continue this process until all nodes in $\mathcal{I}$ are removed. The order in which nodes are removed forms the serialization of the solution. This procedure is detailed in Algorithm 1.

To evaluate the algorithms, we compute the percentage of rounds in which the smallest degree node is selected during serialization, similar to our comparison between `LTFT` and `Deg-Greedy`. Due to random tie-breaking in Algorithm 1, we repeat the process 100 times and select the serialization with the highest number of rounds selecting the smallest degree node. Although `Deg-Greedy` theoretically achieves 100% smallest-degree selections in its best serialization, random tie-breaking prevents us from perfectly recovering this with 100 repetitions. Instead of reporting the overall percentage, we divide the serialization into three equal parts and report the percentages for each part.

The results, shown in Figure 2, compare the percentage of smallest-degree node selections across different algorithms. Due to space constraints, we present results only for ER and BA graphs under selected parameters; additional results are available in Appendix C.2.

Algorithms that often perform the best, namely `OnlineMIS`, `ReduMIS`, and `iSCO`, exhibit a consistent pattern after serialization. In the first one-third of the serialization, these algorithms deviate significantly from selecting the smallest-degree nodes. However, in the middle and final thirds, the percentage of smallest-degree node selections increases substantially. This suggests that while degree-based greedy heuristics may appear shortsighted initially, they are highly effective in the later stages of solution construction. Interestingly, `LwD`, which performs the best among learning-based methods tested in our setting, also shares this pattern.

As for `PCQO` and `DIFUSCO`, they show consistently low percentages of smallest-degree node selections throughout the serialization, particularly on ER graphs.

Through serialization, we observe distinct differences in node selection patterns among algorithms. Our findings suggest that AI-based methods might fail to utilize (`PCQO` and `DIFUSCO`) or emphasize too much (`LTFT`) on simple yet highly effective heuristics, such as greedily selecting the smallest-degree node, which may partly explain their performance limitations.

In Appendix C.3, we perform a *pseudo-natural serialization* for `LwD`. `LwD` is also a non-backtracking algorithm like `LTFT`, but it does not have a "natural serialization" like `LTFT` (Section 4.1) because it chooses several nodes in a step. The *pseudo-natural serialization* performs serialization in each step of `LwD`. The results in Figure 6 align with our "counterfactual" serialization results here.

### 4.3 Incorporating local search to improve solution

In the previous sections, we show that solutions generated by AI-based algorithms generally differ from those produced by degree-based greedy methods, which may explain their inferior performance on MIS

problems. A natural idea is to enhance these solutions with simple heuristics, such as local search. Local search post-processing has also been used for AI-algorithms in previous works (Ahn et al., 2020; Böther et al., 2022).

We applied the 2-improvement local search (Andrade et al., 2012) (details in B.7), which is used in `KaMIS`, as a post-processing step to all algorithms (except `OnlineMIS` and `ReduMIS` since they already has local search), and the resulting performance improvements are presented in Table 6.

Table 6: **Incorporating local search as a post-processing procedure on selected ER graphs.** We add 2-improvement local search for the algorithm tested. We report the performance after the local search and the improvement from local search in (). Full results for ER and BA graphs in Table 10.

| | Heuristics | | | GPU-acc | | Learning-based | | |
|---|---|---|---|---|---|---|---|---|
| $(n, d)$ | Deg-Greedy | OnlineMIS | ReduMIS | iSCO | PCQO | LwD | LTFT | DIFUSCO |
| 1000,30 | 152.75 (1.75) | 158.88 | 163.75 | 163.50 (0) | 159.13 (0.5) | 158.62 (0.24) | 151.38 (1.38) | 150.62 (6.87) |
| 1000,100 | 60.75 (0.13) | 64.75 | 66.63 | 66.50 (0) | 60.88 (0.75) | 64.12 (0.24) | 61.88 (1.00) | 57.88 (2.50) |
| 3000,30 | 456.12 (4.24) | 480.88 | 493.13 | 491.62 (0) | 469.75 (5.5) | 474.00 (0.75) | 454.38 (5.38) | 442.75 (29.37) |

As shown in Table 6, algorithms like `PCQO` and `DIFUSCO` benefit significantly more from the local search post-processing compared to others, such as `Deg-Greedy` and `LwD`. This observation aligns with our earlier findings: If the solution after serialization exhibits a high percentage of smallest-degree node selections in the later stages, there is relatively little room for improvement through local search. Conversely, if the solution after serialization shows a low percentage of smallest-degree node selections, there is greater potential for improvement via local search.

In addition, although all algorithms except `iSCO` have improvements after local search, they still perform worse than `ReduMIS` in most cases.

In summary, our analysis highlights a promising direction for designing machine learning-based combinatorial optimization algorithms. Rather than relying solely on end-to-end methods like `PCQO` or `DIFUSCO`, incorporating classical heuristics, such as greedily selecting the smallest-degree node, into the overall algorithm may yield better results. One potential approach could involve using machine learning algorithms to identify a small subset of nodes, followed by a degree-based greedy method to complete the solution.

### 4.4 Perspective for theoreticians: Empirical performance vs asymptotic conjecture

For theoretical analysis for MIS on ER graphs and regular graphs, see Coja-Oghlan & Efthymiou (2015); Wormald et al. (1999); Barbier et al. (2013); Gamarnik & Sudan (2014). In ER graphs with $n$ nodes and average degree $d$, the MIS has size $\frac{2n \ln d}{d}$ for asymptotically large $n$ and $d$, and simplest random greedy achieves half-optimal at $\frac{n \ln d}{d}$ (Grimmett & McDiarmid, 1975). Yet, there is no known polynomial-time algorithm which can achieve MIS size $(1+\epsilon)\frac{n \ln d}{d}$ for any constant $\epsilon$ and it is conjectured that polynomial-time algorithms cannot do better than $(1 + o(1))\frac{n \ln d}{d}$ (Coja-Oghlan & Efthymiou, 2015). Thus it is natural to measure the goodness of an algorithm by the ratio of the MIS size obtained to $\frac{n \ln d}{d}$. This ratio can faciliate comparison across different $n$ and $d$'s.

Figure 3 plots this ratio for several algorithms. (Plots for all algorithms in Appendix C.5.) Surprisingly for `ReduMIS` and `OnlineMIS` this ratio is consistently larger than 1.2 and often larger than 1.3 even for fairly large $n, d$. One possible reasoning for our finding is that the graphs we are able to evaluate empirically are very far from the asymptotic regime where the conjecture might be applicable (i.e., if it is true). We note that the proofs for Theorem 2 in the original paper had to assume $d > \exp(20)$ and thus $n > \exp(40)$, much larger than any practical sizes of graphs. Our findings suggest that indeed the conjecture itself may not hold for graphs one may encounter in real life. Of course this does not disprove the conjecture per se since it concerns asymptotically large $n$ and $d$, but our results could encourage further analysis and potential collaboration between researchers on theoretical and empirical aspects of MIS on random graphs.

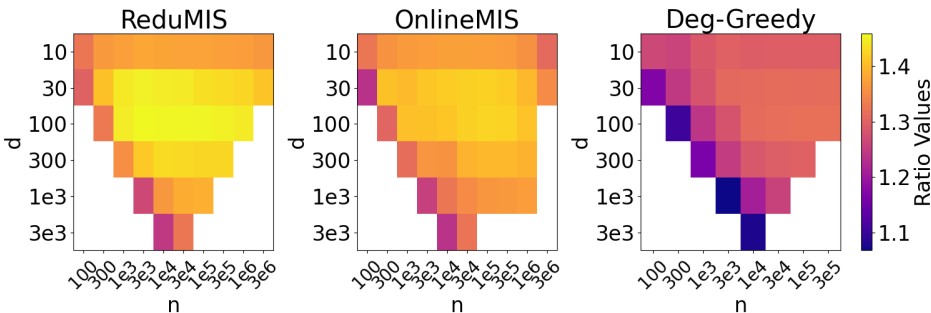

Figure 3: Heatmap for ratios of MIS size to $\frac{n \ln d}{d}$ on ER graphs. We find that the ratio for `ReduMIS` and `KaMIS` is consistently larger than 1.2, suggesting that `ReduMIS` and `KaMIS` might surpass the conjectured upper bound (Coja-Oghlan & Efthymiou, 2015).

## 5 Related Works

**Classical and heuristic methods for MIS**  Classical methods for MIS range from simple greedy algorithms to advanced solvers like `KaMIS` which involves a number of heuristics. There are various existing heuristics for MIS, such as reduction techniques (Butenko et al., 2002; Xiao & Nagamochi, 2013; Akiba & Iwata, 2016), local search (Andrade et al., 2012), and evolutionary algorithms (Back & Khuri, 1994; Borisovsky & Zavolovskaya, 2003; Lamm et al., 2015). `KaMIS` (Lamm et al., 2017; Dahlum et al., 2016) was developed based on these heuristics. In addition, MIS can be formulated into a binary integer programming problem (Nemhauser & Trotter Jr, 1975), which can be solved by the state-of-art integer programming solver `Gurobi` Gurobi Optimization, LLC (2024).

**Machine learning for combinatorial optimization**  In recent years, various ML-based algorithms have been developed for the MIS problem and most of them are based on graph neural networks (GNNs). Some of them using supervised learning (Li et al., 2018; Sun & Yang, 2023; Li et al., 2024) and requires labelling training data using classical solvers. Alternatively, those based on reinforcement learning (Khalil et al., 2017; Ahn et al., 2020; Qiu et al., 2022; Sanokowski et al., 2023) and other unsupervised learning objectives (Karalias & Loukas, 2020; Sun et al., 2022; Zhang et al., 2023; Sanokowski et al., 2024) do not require labeled training data. Zhang et al. (2023) (`LTFT`) uses GFlowNets (Bengio et al., 2021a) which is related to reinforcement learning. Notably, Ahn et al. (2020); Sanokowski et al. (2023); Zhang et al. (2023) model MIS problem as a Markov Decision Process (MDP) and generate the solution step-by-step (autoregressively). While Sanokowski et al. (2023) fixes the order of node updates, Ahn et al. (2020) (`LwD`) and Zhang et al. (2023) (`LTFT`) choose which node to update at each step so that they have a "natural" or "pseudo-natural" serialization and most suitable for our analysis in Section 4.1 and Appendix C.3. In addition, Sun & Yang (2023) (`DIFUSCO`) and Sanokowski et al. (2024) (`DiffUCO`) both utilizes diffusion model.

Most of the algorithms above also work on other types of graph CO problems, such as Maximum Cut, Maximum Vetex Cover, and Minimum Dominating Set. Some (Khalil et al., 2017; Qiu et al., 2022; Sun & Yang, 2023) also work for the Travelling Salesman Problem (TSP).

**Other GPU-based solvers for CO**  Recently, some non-learning GPU-based solvers for CO problems have been developed. Sun et al. (2023) developed a GPU-accelerated sampling based method which works on MIS, Max Cut, and TSP. Schuetz et al. (2022); Ichikawa (2023) uses GNNs conduct non-convex optimization for MIS without machine learning. Alkhouri et al. (2025) uses direct quadratic optimization without GNNs.

**Benchmarks for MIS**  Böther et al. (2022) provides a benchmark for several MIS algorithms including `Gurobi` (Gurobi Optimization, LLC, 2024), `KaMIS` (`ReduMIS`) (Lamm et al., 2017), Intel-Treesearch (Li et al., 2018), DGL-Treesearch, and `LwD` Ahn et al. (2020). It is the only MIS benchmark we know about including recent AI-inspired method, though it only focuses its comparsion for learning-based tree search algorithms. It suggests that `LwD` is better than learning-based tree search algorithms. This aligns with our results where `LwD`

performs the best among learning-based algorithms. This benchmark covers various types of random graphs and several real-world datasets, so it is a good reference benchmark for comparison over different types of graphs. Though unlike our benchmark, it does not provide comparison across various size and density for random graphs. Our benchmark fills this gap and provides a more detailed comparison. We also include many newer AI-inspired algorithms, and greedy algorithms which leads to detailed analysis like serialization.

Angelini & Ricci-Tersenghi (2023) showed that one specific GNN-based MIS algorithm (Schuetz et al., 2022) failed to surpass `Deg-Greedy` on regular graphs. Although their title ("Modern graph neural networks do worse...") appears to suggest a broad conclusion about GNN-based methods, their empirical evaluation was limited to that specific algorithm. The algorithm papers also report experiments comparing with previous algorithms, but those comparisons usually only focus on a few datasets and a few selected baselines.

## 6 Conclusion and Takeaways

Given the great interest in designing "general purpose AI reasoners", it is interesting to check how well recent AI-based methods have fared in combinatorial optimization, a field with a long history of ingenious hand-designed algorithms. Our careful empirical comparisons on the MIS problem showed that none of the new AI-inspired methods outperform `ReduMIS`, the best CPU-based MIS solver. Strikingly, this underperformance holds even on *in-distribution random graphs*, a setting arguably ideal for machine learning-based approaches. As the graphs get larger or denser, the superiority of `ReduMIS` becomes more evident, whereas several AI-inspired algorithms degrade to performing no better than the simple degree-based greedy heuristic (`Deg-Greedy`).

A key contribution of this work is the introduction of *serialization*, a novel analysis technique to deconstruct and compare the decision-making processes of different algorithms. This analysis revealed that some AI-inspired methods, such as the GFlowNet-based `LTFT`, end up reasoning very similarly to the simple `Deg-Greedy` heuristic, which helps explain their performance limitations. More importantly, serialization uncovered a distinct pattern shared by high-performing algorithms like `OnlineMIS`, `ReduMIS`, `iSCO`, and `LwD`: they deviate from simple greedy choices in the initial stages but align with them in the later stages of constructing a solution. This suggests serialization is a powerful tool for discovering the intrinsic characteristics of high-quality solutions, which can provide overlooked insights for the principled design of future algorithms.

Serialization-based analysis also shows the importance of algorithmic strategies in understanding this performance gap. One-shot methods like `DIFUSCO` and `PCQO` handicap themselves by foregoing the benefits of *local search*, whereas non-backtracking methods like `Deg-Greedy` and `LTFT` handicap themselves by never deleting vertices from the independent set being built. In contrast, `KaMIS` performs a full local search, allowing it to iteratively add and delete vertices. Interestingly, the best-performing learning-based algorithm, `LwD`, operates in a middle ground by selecting several nodes at a time, which may allow a closer approximation to local search. And while post-processing with local search improves the solutions of most AI-inspired methods, they still fail to match the performance of `ReduMIS`.

While prior work has questioned whether AI-inspired methods consistently outperform strong classical baselines, our evaluation of more recent approaches suggests that these challenges persist at practically relevant scales. Similar patterns have been observed in the Traveling Salesperson Problem (TSP), where neural methods often struggle to surpass specialized classical solvers under careful benchmarking (Joshi et al., 2022; Xia et al., 2024). Notably, both MIS and TSP are among the most extensively studied problems in AI-based combinatorial optimization, yet in both cases claimed advantages of AI-based methods weaken under more extensive and carefully designed evaluation. This suggests that rigorous and comprehensive benchmarking is essential across combinatorial optimization problems when assessing AI-based approaches.

Moreover, while our study focuses on MIS, the lesson of principled integration of classical heuristics may extend to other problems—particularly closely related graph problems such as Maximum Clique, Vertex Cover, and Graph Coloring, which share similar structural and scalability challenges.

We thus propose that future work should prioritize the following directions:

- **Rigorous and Comprehensive Benchmarking:** Future work must establish more realistic benchmarks that evaluate algorithms on a wider range of instance sizes and densities. They are necessary for meaningful comparisons and provide important guidance for future developments. The evaluations should also report crucial computational costs, such as GPU memory usage, to ensure fair comparisons against efficient CPU-based solvers that are often more scalable in practice.

- **Deeper Understanding of Learning Limitations:** A concerted effort is needed to understand theoretical barriers like Gamarnik (2023) for GNN-based methods. Meanwhile, expanding on empirical tools like *serialization* to pinpoint specific failure modes is also important for guiding the development of new algorithms.

- **Principled Integration of Classical Heuristics:** Researchers should move beyond treating classical solvers as black-box baselines and instead analyze, learn from, and integrate their most effective components. A promising direction is to design hybrid models that use AI to guide classical heuristics—for instance, by using a learned model to identify a subset of nodes and select suitable heuristics (such as reduction, local search, or degree-based greedy) to apply. Previous work by Garmendia et al. (2023) demonstrates the potential of such neural improvement heuristics, and we advocate for further exploration in this direction.

### Acknowledgments

YW, HZ, and SA acknowledge funding from DARPA. The authors would like to thank Alvaro Velasquez, Ismail Alkhouri, Kaifeng Lyu, Sadhika Malladi, and Pravesh Kothari for helpful discussions.

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

## Appendices

## A  More Related Works

**More on theoretical results for MIS**   For random ER graphs with number of nodes $n$ and average degree $d$, the upper bound of MIS size is $\frac{2n \ln d}{d}$ for asymptotically large $n$ and $d$. (Coja-Oghlan & Efthymiou, 2015). Grimmett & McDiarmid (1975) proves that the simplest `Ran-Greedy` can achieve half-optimal at $\frac{n \ln d}{d}$ on random ER graphs. Despite that, there is no known existing polynomial algorithm which can reach MIS size of $(1 + \epsilon)\frac{n \ln d}{d}$ for any constant $\epsilon$ for asymptotically large $n$ and $d$. (Coja-Oghlan & Efthymiou, 2015). Coja-Oghlan & Efthymiou (2015) also suggests that the reason is likely independent sets of size larger than $(1 + \epsilon)\frac{n \ln d}{d}$ forms an intricately ragged landscape, where local algorithms will stuck. Gamarnik & Sudan (2014); Rahman & Virág (2017) proves that for local algorithms (which is defined to only use information from a constant neighborhood of a node to decide whether the node is in the independent set) are at most half optimal for independent set on random $d$-regular graphs. Gamarnik & Sudan (2014) suggests this is due to a property of the MIS problem, which they denote as the *Overlap Gap Property* (OGP) (Gamarnik, 2021). Gamarnik (2023) suggests that graph neural networks (GNNs) are also a type of local algorithms and thus being limited by OGP. While most AI-inspired MIS algorithms use GNN, the proof only applies to algorithms use GNNs as the only component to find solutions like (Schuetz et al., 2022), so it may not apply directly to more complicated algorithms like those tested in our paper. Yet, it may still suggests a reason

why GNN-based algorithms (including most AI-inspired algorithms) cannot outperform classical heuristics like `KaMIS`.

In addition, Barbier et al. (2013) provides a conjectured tighter upper bound than $\frac{2n \ln d}{d}$ for $d$-regular graphs using the hard-core model in physics. Ding et al. (2016) proves a similar tighter upper bound for $d$-regular graphs. Wormald (2003) gives average-case performance for `Deg-Greedy` on $d$-regular graphs.

**More on classical heuristics**   Over the past few decades, significant progress has been made in tackling NP-hard combinatorial optimization (CO) problems by developing approximation algorithms and heuristic methods. Approximation algorithms provide provable guarantees on solution quality and have led to groundbreaking results for classical problems, such as the Maximum Independent Set (MIS), Traveling Salesperson Problem (TSP), and Maximum Cut (Boppana & Halldórsson, 1992; Laporte, 1992; Goemans & Williamson, 1995).

As we mentioned in Section 5, there are various existing heuristics for MIS. For exmaple, reduction techniques reduce the graph into smaller instances. Akiba & Iwata (2016) and Xiao & Nagamochi (2013) have shown a variety of reduction techniques which work well for MIS problem. Butenko et al. (2002); Bourgeois et al. (2012) uses reduction techniques to develop efficient exact algorithms for MIS. Local search improves an existing independent set by removing a small number of nodes and insert other eligible nodes. Andrade et al. (2012) gives an efficient local search algorithm for MIS and has been used as a subprocess for several MIS solvers. Evolutionary algorithms combine several existing solutions into a new solution. Examples include Back & Khuri (1994); Borisovsky & Zavolovskaya (2003); Lamm et al. (2015). `KaMIS` (Lamm et al., 2017; Dahlum et al., 2016) was developed based on many of these techniques above.

In addition, the MIS problem can also be relaxed into semi-definite programming (SDP), which leads to several approximation algorithms (Halperin, 2002; Bansal, 2014).

**More on AI methods for combinatorial optimization**   In addition to Section 5, we note that Sun et al. (2022); Sanokowski et al. (2023; 2024) use annealing techniques. Sun & Yang (2023) was developed based on Qiu et al. (2022), and Sanokowski et al. (2024) was based on Sanokowski et al. (2023). Sanokowski et al. (2024) can also be considered as an extension of Karalias & Loukas (2020).

Besides MIS, Maximum Cut, and TSP, people also considered to use AI-Inspired methods for other combinatorial optimization problems, including Vehicle Routing Problems (including TSP) (Kool et al., 2018; Chen & Tian, 2019; Delarue et al., 2020; Li et al., 2021; Zheng et al., 2021; Ye et al., 2024), Job Scheduling Problems (Lin et al., 2019; Baer et al., 2019; Zhang et al., 2020; Ye et al., 2024), Boolean Satisfiability (SAT) (Amizadeh et al., 2019; You et al., 2019; Kurin et al., 2020; Li & Si, 2022; Li et al., 2023), and Casual Discovery (Zheng et al., 2018; Zhu et al., 2020; Sanchez et al., 2022).

## B  Detailed Experiment Setup

### B.1  Algorithm Pseudo-code for degree-based serialization

Please refer to Algorithm 1 for the pseudo-code for degree-based serialization, mentioned in Section 4.2.

### B.2  Datasets

For synthetic graphs, we test 8 graphs for each parameter $(n, d)$ or $(n, m)$. We test on 100 graphs for real-world datasets. For learning-based algorithms, we use 4000 training graphs generated using the same parameter (in case of random graphs) or drawn 4000 graphs from the same real-world dataset.

**Erdős-Renyí (ER) graph**   ER graphs (Erdös & Rényi, 1959) are random graphs where edges are connected uniformly at random (with a fixed probability or number of edges). We vary 2 parameters for ER graphs, number of nodes $n$ and average degree $d$, by fixing number of edges at $\frac{nd}{2}$. ER graphs are fundamental objects in network sciences (Lewis, 2011) and random graph theory (Bollbás & Bollobás, 1998). There are also theoretical analysis and conjecture upper bound for MIS on ER graphs (Coja-Oghlan & Efthymiou, 2015).

---

**Algorithm 1** Degree-based solution serialization

---

**Require:** The graph $\mathcal{G}(\mathcal{V}, \mathcal{E})$, the independent set $\mathcal{I}$
 1: // Maintain the copy
 2: $\mathcal{G}' \leftarrow \mathcal{G}, \mathcal{I}' \leftarrow \mathcal{I}$
 3: // Initialize empty ordered list
 4: $\mathcal{L} = []$
 5: **while** $\mathcal{I}' \neq \phi$ **do**
 6:     pick $v \in \mathcal{I}'$ with smallest degree in $\mathcal{G}'$, break ties randomly
 7:     $\mathcal{L} \leftarrow \mathcal{L} \cup \{v\}$
 8:     Delete $\{v\} \cup \texttt{Neighbors}(v)$ from $\mathcal{G}'$, delete $v$ from $\mathcal{I}'$
 9: **end while**
10: **return**  Ordered list $\mathcal{L}$

---

Previous works (Böther et al., 2022; Ahn et al., 2020; Sun & Yang, 2023; Zhang et al., 2023; Alkhouri et al., 2025) used it as test graphs for MIS, though without varying parameters as we did.

ER graphs have 2 parameters: the number of nodes $n$ and the average degree $d$. For graphs with given $(n, d)$ We generate them by choosing $M = \frac{nd}{2}$ edges uniformly at random between $n$ nodes. This is the Erdős-Renyí's $G(n, M)$ model.

There is also $G(n, p)$ model for ER graphs, which is also used widely. They behave similarly for many graph properties to $G(n, M)$ models when $M = \binom{n}{2}p$ and we expect the emipirical results for MIS problems will also be very similar.

For the main experiments, we use ER graphs with $n = \{100, 300, 1000, 3000, 10000\}, d = \{10, 30, 100, 300, 1000, 3000\}$ with $d < n$ as shown in Table 3. We also test on larger graphs with $n = \{30000, 1 \times 10^5, 3 \times 10^5, 1 \times 10^6, 3 \times 10^6\}$ as shown in Table 9. Due to computational and time limits, we could only obtain results from classical CPU-based algorithms for these large graph, and only sparse graphs for very large $n \geq 1 \times 10^6$.

**Barabási–Albert (BA) graph**  Barabási–Albert (BA) graphs (Albert & Barabási, 2002) are random graphs generated by a probabilistic growth process, whereby new nodes preferentially attach to existing nodes with higher degrees, mimicking real-world networks such as Internet, citation networks, and social networks (Albert & Barabási, 2002; Radicchi et al., 2011). For BA graphs, we vary 2 parameters: number of nodes $n$ and parameter $m$ (not number of edges). The average degree of BA graphs can be approximated as $2m$.

BA graphs also have 2 parameters: the number of nodes $n$ and the generation parameter $m$, with $n > m$. For a given $(n, m)$, the generation process initializes with $m$ nodes, and then add 1 node at each step. When adding a new node, $m$ neighbors of the new node are sampled from the existing nodes, with the probability of the current degree of the nodes. The average degree of BA graph with given $(n, m)$ can be computed as $d = 2m - \frac{m}{n} - \frac{m^2}{n} \approx 2m$.

For the main experiments, similar to ER graphs, we also use $n = \{100, 300, 1000, 3000, 10000\}$. Since the average degree $d \approx 2m$, we use $m = \{5, 15, 50, 150, 500, 1500\}$ with $2m < n$.

**RB graphs**  RB graphs are derived from Model RB (Xu & Li, 2000), a random constraint satisfaction problem (CSP) model. RB graphs are considered difficult instances for MIS due to their structured randomness and high solution hardness.

We use two datasets from Zhang et al. (2023) (also used in Sanokowski et al. (2024)), `RB-small` (200–300 nodes) and `RB-large` (800–1200 nodes), to benchmark learning-based solvers.

**Real-world graphs**  We pick REDDIT-MULTI-5K and COLLAB (Yanardag & Vishwanathan, 2015) from TUDataset website (Morris et al., 2020), since they have enough graphs for training and graph sizes not too small. REDDIT-MULTI-5K has 508.52 average nodes and 594.87 average edges. They are mostly very sparse

graphs. COLLAB has 74.49 average nodes and 2457.78 average edges. They are mostly small but dense graphs.

Since we need at least 4100 graphs to train and test our algorithms and the graphs should not be too small, it is difficult to find such datasets. Fortunately, Morris et al. (2020) provides a website `www.graphlearning.io` which includes many graph datasets prepared by them or collected from other works. Although most of the datasets are still too small or having too small graphs, we are able to find 2 datasets: REDDIT-MULTI-5K and COLLA, both from Yanardag & Vishwanathan (2015). REDDIT-MULTI-5K has 4999 graphs with average nodes 508.52 and average edges 594.87, so the graphs are generally very sparse. COLLA has 5000 graphs with average nodes 74.49 and average edges 2457.78, so they are denser but smaller graphs.

We were not able to find real-world datasets with enough size of more larger and denser graphs, which are generally more difficult for MIS algorithms. The dataset DIMACS used in Böther et al. (2022) contains such graphs but they only have 37 graphs which is not enough for training.

## B.3    Hardware configurations

The CPU we use is either Intel Xeon Processor E5-2680 v4 @ 2.40GHz or Intel Xeon Silver 4214 Processor @ 2.20GHz. The GPU we use is Nvidia Tesla A100 80GB when we refer to time limit or time cost. For small graphs when the GPU memory limit and time limit is not reached, we also use Nvidia RTX A6000 48GB and Nvidia RTX 2080Ti 11GB.

## B.4    Classical CPU-based algorithms

**Ran-Greedy and Deg-Greedy**   `Deg-Greedy` (Degree-based Greedy) is as follows: Starting from an empty set. Select a node with the lowest degree of the graph (if there exists several nodes of the lowest degree, we pick one uniformly at random) and add it to the set. Then remove the node and all its neighbors from the graph.

`Ran-Greedy` (Random Greedy) is as follows: Starting from an empty set. Select a node from the graph uniformly at random and add it into the set. Then remove the node and all its neighbors from the graph. The only difference between it and `Deg-Greedy` is that the choice of node is completely random.

They are both non-backtracking algorithms.

We did not include results of `Ran-Greedy` in the main paper because its performance is significantly lower than all other algorithms. It can be considered as a baseline and has theoretical significance, since it has provable guarantee for random graphs (Grimmett & McDiarmid, 1975).

In order to match the best-of-20 sampling we used for the learning-based-algorithms, we also ran `Deg-Greedy` for 20 times and report the best results in the main experiments ($n \leq 10000$).

We wrote the script in Python and ran it on a single CPU thread for each graph with a time limit of 24hrs, but it actually run less than 1hr on smaller graphs like $n \leq 3000$. We gave 32GB memory for graphs with $n \leq 10000$ and 64GB memory for larger graphs. Since there are much more efficient implementations like C++, the efficiency of the greedy algorithms is not very relevant.

**KaMIS (OnlineMIS and ReduMIS)**   `KaMIS` (Karlsruhe Maximum Independent Sets) (Lamm et al., 2017) (`https://karlsruhemis.github.io/`) is the state-of-art heuristic solver for MIS and has been used as a baseline in many previous works. It provides 2 algorithms for the MIS problem: `ReduMIS` (Lamm et al., 2017) and `OnlineMIS` (Dahlum et al., 2016).

We can provide a time-limit for both algorithm. `OnlineMIS` will use up the time given while `ReduMIS` will end on its own when it finds appropriate. In general, `ReduMIS` provides better results when given enough time, where `OnlineMIS` is faster to reach a solution of reasonable quality for large and dense graphs.

We ran both algorithm on a single CPU thread for each graph with a time limit of 24hrs, because our benchmark focus on performance instead of efficiency. For relatively small graphs ($n \leq 3000$), `ReduMIS` often require less than 1hr and at most 1.25hrs, and `OnlineMIS` can also provide answer with the same quality

when giving 1hr time limit. We gave 32GB memory for graphs with $n \leq 10000$ and 64GB memory for larger graphs.

## B.5 GPU-accelerated non-learning algorithms

**iSCO** iSCO (improved Sampling for Combinatorial Optimization) (Sun et al., 2023) is a GPU-accelerated sampling-based method. It does not require learning. According to (Sun et al., 2023), the main benefit is that it can process a large batch of graphs in parallel thus improve efficiency. While processing a small number of graphs like in our case (8 test graphs), it still requires significant time, often longer than `ReduMIS`.

We use the code from the codebase of DISCS (Goshvadi et al., 2024), which is a follow-up paper for `iSCO`, as `iSCO` did not provide the codebase. We use 1 80GB $A$100 GPU to run `iSCO` with all test graphs together. The time limit is 96hrs, and the actual time it requires is shorter. It fail to run graphs of size $(n = 3000, 1000)$, $(n = 10000, d = 100)$, and larger because they require larger than 80GB memory. The code does not support multi-GPU.

**PCQO** PCQO (Parallelized Clique-Informed Quadratic Optimization) (Alkhouri et al., 2025) is a GPU-accelerated gradient-based optimization algorithm, which directly optimize on the quadratic loss function of the MIS problem. The loss function for a graph $G = (V, E)$ and solution vector $\boldsymbol{x}$ is

$$f(\boldsymbol{x}) \coloneqq -\sum_{v \in V} \boldsymbol{x}_v + \gamma \sum_{(u,v) \in E} \boldsymbol{x}_v \boldsymbol{x}_u - \gamma' \sum_{(u,v) \in E'} \boldsymbol{x}_v \boldsymbol{x}_u, \tag{1}$$

where $E'$ is the edge set of the complement graph $G'$. $\gamma$ and $\gamma'$ are hyperparameters and there are many other hyperparameters including learning rate, momentum, etc.

This algorithm is sensitive to hyperparameters and the default hyperparameters lead to bad solution quality for most of our dataset. Therefore, unlike in other algorithms where we use the default setting, we perform a hyperparameter tunining. We also did not find a good set of hyperparameters for all ER graphs or all BA graphs, so we do a grid search of hyperparameters for each dataset (i.e. for each $(n, d)$ pair in ER graphs and each $(n, m)$ pair in BA graphs).

We use the hyperparamter search range suggested by the authors (Alkhouri et al., 2025). We use learning rate $\{0.0001, 0.00001, 0.000001\}$, momentum 0.9, $\gamma \in \{200, 500, 1000, 2000, 5000\}$, $\gamma' = 1$, batch size 256, number of steps 27000, steps per batch 450. We then report the results obtained from the best set of hyperparameter. However, for $n \geq 30000$, the grid search within this domain does not provide solution with reasonable size (worse than `Ran-Greedy`), so we do not report results for larger graphs. Despite that, there may exist a better set of hyperparamters which could make this algorithm perform well on larger graphs. Automatic hyperparameter search could significantly improve the usability of this algorithm.

We use 1 80GB $A$100 GPU for all test graphs together with a time limit of 96hrs. It is able to run all experiments from $n \leq 10000$, and often requires shorter time compared to `iSCO` and `KaMIS`.

## B.6 Learning-based algorithms

We use 4000 training graphs for each datasets to train our models. Without otherwise noted, all the training are in-distribution, with respect to a single set of parameters (i.e. $(n, d)$ for ER graphs, $(n, m)$ for BA graphs) for synthetic graphs.

**LwD** LwD (Learning what to Defer) (Ahn et al., 2020) is a reinforcement learning based algorithm which requires training data to learn the policy. It models the MIS problem as a Markov Decision Process (MDP), where in each step it selects some (possibly 0, 1, or multiple) nodes to add into the independent set. It is a non-backtracking algorithm as the added nodes are never taken out. Böther et al. (2022) also included this algorithm in their benchmark.

We use the code from Böther et al. (2022) since it provides better functionality than the original codebase. We use the default setting provided by Böther et al. (2022) in their MIS Benchmark codebase (`https:`

//github.com/MaxiBoether/mis-benchmark-framework), but change the number of samples to 20 (default is 10) for test sampling, in order to match the best-of-20 sampling in our benchmark.

We use 1 80GB $A$100 GPU to train for each datasets and test with the same GPU. The training time limit is set to 96hrs. The default number of training steps (number of updates to the policy) is 20000. Since `LwD` stores checkpoints throughout the process, we still report the test results based on the newest checkpoints for unfinished experiments if we have the checkpoints which reports meaningful results (better than half of the results reported by `Deg-Greedy`). Those results are indicated by ∗ in the tables. The number of steps taken by those datasets with unfinished experiments is in Table 7.

Table 7: **Number of Steps at termination for unfinished `LwD` experiments**

| Type of Graphs | Parameters | Number of Steps at termination |
|---|---|---|
| ER $(n, d)$ | $(1000, 300)$ | 1500 |
| | $(3000, 100)$ | 7200 |
| BA $(n, m)$ | $(1000, 50)$ | 18600 |
| | $(1000, 150)$ | 9600 |
| | $(3000, 15)$ | 9300 |
| | $(3000, 50)$ | 600 |

**LTFT**  LTFT (Let the Flows Tell) (Zhang et al., 2023) similar to `LwD`, also model the MIS problem as a MDP and non-backtracking. The difference is that it only choose 1 node at each step, making it more similar to `Deg-Greedy`. The node chosen at each step is chosen by a GFlowNet (Bengio et al., 2021a), which is trained by in-distribution training data.

We use the default setting provide by Zhang et al. (2023) for training with 20 epochs. By default setting, it has best-of-20 sampling and report the best solution found.

We use 1 80GB $A$100 GPU to train for each datasets and test with the same GPU. The training time limit is set to 96hrs. It completes training for all graphs with $n \leq 3000$, but larger graphs require larger GPU memory. The code does not support multi-GPU.

**DIFUSCO**  DIFUSCO (Diffusion Solvers for Combinatorial Optimization) (Sun & Yang, 2023) trains a diffusion model using supervised learning to produce a solution for the MIS. The diffusion model provides an entire solution so it is a one-shot algorithm.

The training data is 4000 graphs for each dataset (1 set of parameter for synthetic graphs). All training is in-distribution. The training data is labelled by `ReduMIS` with time limit of 1hr. For graphs we used for training ($n \leq 3000$), `ReduMIS` gives the same performance compared to a time limit of 24hrs.

We use the default setting in (Sun & Yang, 2023) except that we use 50 diffusion steps throughout training and testing, and 20 samples for testing to be aligned with best-of-20 sampling in other methods. We train the model for 50 epochs (default) for each dataset.

We use 1 80GB $A$100 GPU to train for each datasets and test with the same GPU. The training time limit is set to 96hrs. The code does not support multi-GPU. We report results where the training can be completed.

Although Sun & Yang (2023) suggested that `DIFUSCO` has some generalization ability. We found the performance degrade significantly for out-of-distribution trained models (specifically trained on smaller graphs with the same average degree but test on larger graphs), we did not report the results of larger graphs where the in-distribution training cannot finish.

**DiffUCO**  DiffUCO (Diffusion for Unsupervised Combinatorial Optimization) (Sanokowski et al., 2024) is also a diffusion model based algorithm but unlike `DIFUSCO` it uses unsupervised learning. The diffusion model

is trained to sample the solution of low energy state. It also provides an entire solution and is also a one-shot algorithm.

We use the default setting in Sanokowski et al. (2024) for RB-large MIS task (in their Appendix C.5). During testing, we use conditional expectation with 20 samples to align with best-of-20 sampling in other algorithms. The code supports multi-GPU. We use 4 80GB $A$100 GPU to train for each datasets with time limit 96hrs.

The training time is significantly longer than other learning-based algorithms for the same dataset and it can only complete training up to ER graphs with ($n = 1000, d = 100$) and BA graph with ($n = 1000, d = 50$). According to Sanokowski et al. (2024) it has reasonable generalization ability, and we also found that the performance drop is relatively small if we test larger graphs using models trained with smaller graphs with similar average degree. Therefore, we also report test results using out-of-distribution trained model. The parameters of those datasets and the datasets used to train corresponding models are reported in Table 8. Those results are labelled using † in tables.

Table 8: **Parameters of Test and Training Graphs for out-of-distribution testing in `DiffUCO`**

| Type of Graphs | Parameters of Test Graphs | Parameters of Training Graphs |
|---|---|---|
| ER $(n, d)$ | (1000, 300) | (1000, 100) |
| | (3000, 10) | (1000, 10) |
| | (3000, 30) | (3000, 30) |
| | (3000, 100) | (1000, 100) |
| BA $(n, m)$ | (1000, 150) | (1000, 50) |
| | (3000, 5) | (1000, 5) |
| | (3000, 15) | (1000, 15) |
| | (3000, 50) | (1000, 50) |

### B.7 Local search

Local search is a method to improve a given independent set. It can be used as a post-processing technique, or be used as sub-procedures in more complicated algorithms like `KaMIS`. Andrade et al. (2012) provides an efficient local search algorithm. Part of it is to find 2-improvement, which is the part used as sub-procedure in `KaMIS` (Lamm et al., 2017; Dahlum et al., 2016).

The local search algorithm for 2-improvement for a given independent set $I$ is as follows. This algorithm process every vertex $x \in I$ in turn. First, it temporarily removes $x$ from $I$, creating a new set $S$. We call a vertex a free vertex of $S$ if there is no edge between it and any vertex in $S$. If $S$ has less than two free vertices, stop: there is no 2-improvement involving $x$. Otherwise, for each neighbor $v$ of $x$ that is a free vertex for $S$, insert $v$ into $S$ and check if the new set ($S'$) has a free vertex $w$. If it does, inserting $w$ leads to a 2-improvement; if it does not, remove $v$ from $S'$ (thus restoring $S$) and process the next neighbor of $x$. If no improvement is found, reinsert $x$ into $S$ to turn it back to $I$. Every vertex is scanned $O(1)$ times in this algorithm so it can find a 2-improvement (if there exists) in $O(m)$ time according to Andrade et al. (2012).

We implemented this algorithm in Python and use it as a post-processing for the solutions produced by the algorithm we test.

## C More Experiment Results

In this section, we show more experiment results. Appendix C.1 shows more experiment results on much larger graphs, where the AI-inspired methods cannot handle. Appendix C.2 show the serialization results on more graphs. Appendix C.3 show a more detailed results between `LwD` and `Deg-Greedy`, which applies degree-based serialization as a subprocedure. Appendix C.4 shows the full results when adding local search as a post-processing procedure.

## C.1 Larger graphs

Table 9 reports our results for large ER graphs not reported in Table 2. Within our computation limits as described in Appendix B, we can only obtain results for classical heuristic algorithms (`Ran-Greedy`, `Deg-Greedy`, `OnlineMIS`, `ReduMIS`).

Table 9: **(Comparison of the performance of different algorithms on larger Erdős–Rényi (ER) graphs)** Continuation of Table 2 on ER graphs with number of nodes larger than 100,000 for classical heuristic algorithms. Other algorithms are out of our computational limits for these large graphs.

| | | Heuristics | | | |
|---|---|---|---|---|---|
| $n$ | $d$ | Ran-Greedy | Deg-Greedy | OnlineMIS | ReduMIS |
| | 10 | 7170.50 | 8951.75 | 9486.00 | 9505.38 |
| | 30 | 3429.38 | 4464.12 | 4832.88 | 4914.88 |
| | 100 | 1386.00 | 1817.25 | 1963.12 | 2012.62 |
| 30000 | 300 | 570.50 | 738.00 | 794.75 | 815.50 |
| | 1000 | 205.63 | 260.75 | 279.88 | 287.50 |
| | 3000 | – | – | 106.00 | 106.12 |
| | 10 | 23990.00 | 29856.38 | 31613.38 | 31650.62 |
| | 30 | 11444.38 | 14870.12 | 16128.88 | 16287.50 |
| | 100 | 4615.25 | 6064.50 | 6564.00 | 6702.38 |
| | 300 | 1886.62 | 2470.75 | 2661.50 | 2714.75 |
| 100000 | 1000 | – | 83.50 | 941.38 | 959.75 |
| | 10 | 71954.38 | 89487.12 | 94622.88 | 94799.75 |
| | 30 | 34318.88 | 44653.38 | 48234.38 | 48713.00 |
| | 100 | 13850.12 | 18190.38 | 19676.50 | 20061.38 |
| 300000 | 300 | 5688.00 | – | 7987.50 | 8141.25 |
| | 1000 | – | 2831.38 | – | – |
| | 3000 | 802.00 | – | – | – |
| | 10 | 239749.12 | – | 312462.62 | 315630.88 |
| | 30 | 114397.00 | – | 158625.29 | 161622.75 |
| 1000000 | 100 | 46161.12 | – | 64915.00 | 66539.75 |
| | 300 | – | – | 26458.75 | – |
| | 1000 | 6890.00 | – | 9473.00 | – |
| 3000000 | 10 | 719348.75 | – | 904613.12 | 942475.57 |
| | 30 | 343479.38 | – | 459671.29 | 479938.75 |

## C.2 Serialization

Figures 4 and 5 shows the percentage to choose the smallest possible degree node on different part of the serialization across various algorithms for all ER graphs and all BA graphs with nodes $n \leq 3000$, respectively. The serialization process is discussed in Section 4.2. Missing bars are algorithms which we do not get results due to computational limit, same as in Tables 2 and 3.

These results reinforced our observations in Section 4.2. First, the percentage for `Deg-Greedy` and `LTFT` are generally high. `Deg-Greedy` reaches 100% for all parts in some graphs, which is the theoretically achievable percentage since `Deg-Greedy` actually picks the lowest degree node in the remaining graph and this sequence

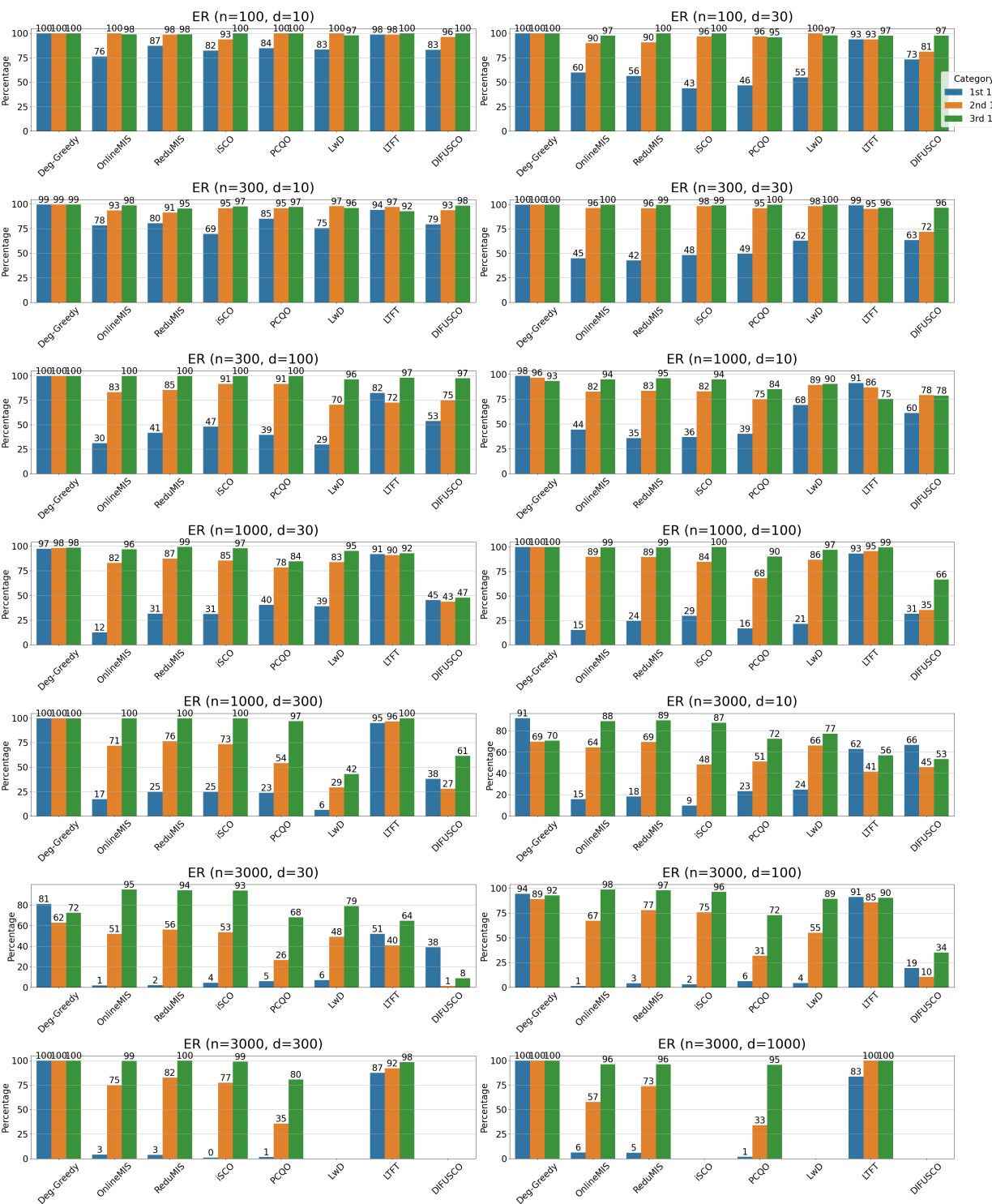

Figure 4: **The percentage to choose the smallest possible degree node on different part of the serialization for all ER graphs** It reinforces our observations in Section 4.2. In addition, we also observe that algorithms similar to `Deg-Greedy` (`Deg-Greedy` and `LTFT`) and good-performing algorithms (`OnlineMIS`, `ReduMIS`, `iSCO`, and `LwD`) have clearly different characteristics across various $(n, d)$, described in Appendix C.2.

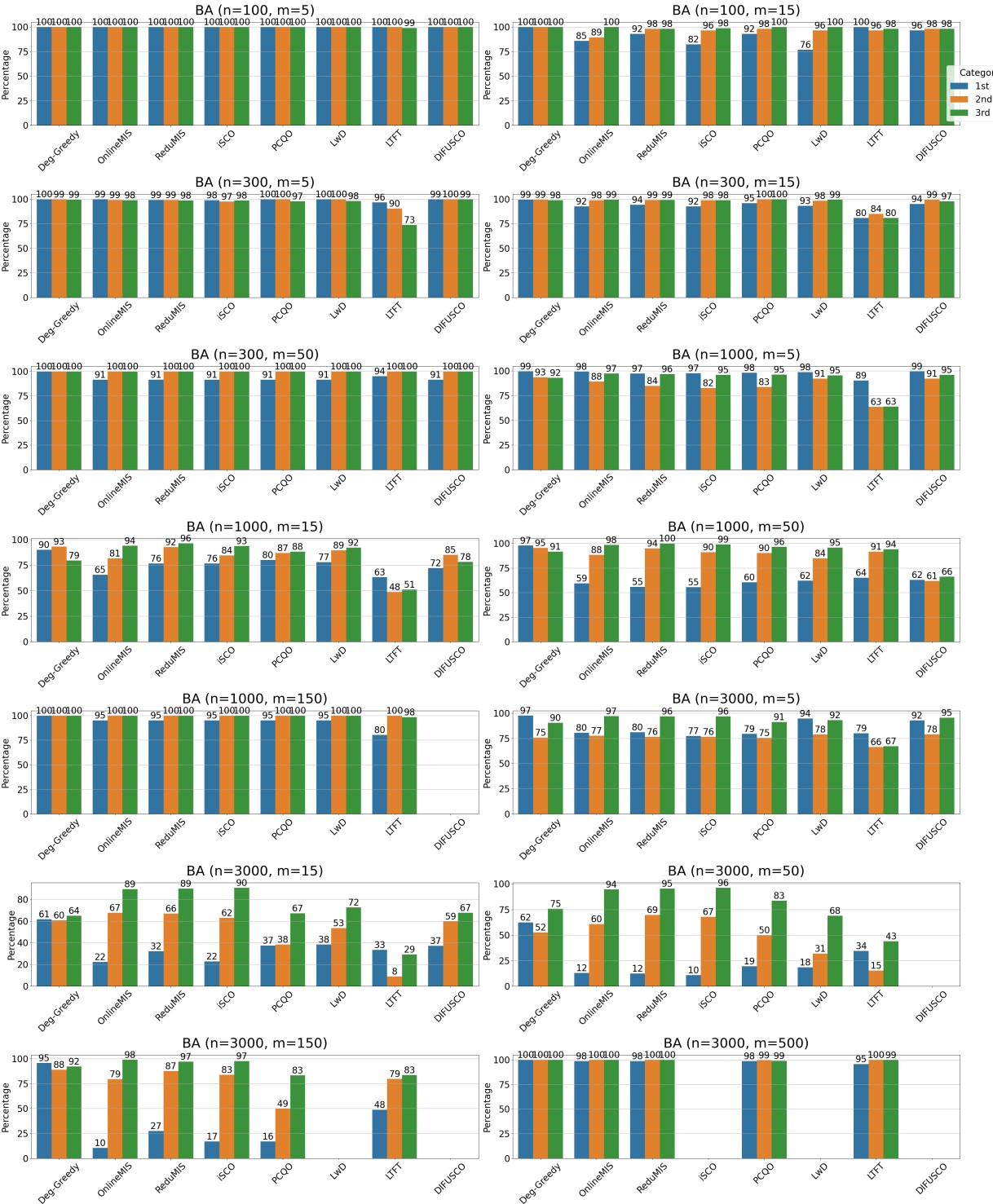

Figure 5: **The percentage to choose the smallest possible degree node on different part of the serialization for all BA graphs** It reinforces our observations in Section 4.2. In addition, we also observe that algorithms similar to `Deg-Greedy` (`Deg-Greedy` and `LTFT`) and good-performing algorithms (`OnlineMIS`, `ReduMIS`, `iSCO`, and `LwD`) have clearly different characteristics across various $(n, m)$, described in Appendix C.2.

will give a serialization with percentage 100% for all parts. In those cases, `LTFT` also have percentage close to 100% for all 3 parts.

Second, for those algorithms with good performance, namely `OnlineMIS`, `ReduMIS`, `iSCO`, and `LwD`, the bar plot shows similar characteristics. The percentage for the 1st third is generally low, while the 2nd third is high, and the 3rd third is higher and close to 100%. This characteristics are observed in most settings accross various parameters $(n, d)/(n, m)$ for both ER and BA graphs. The exception is only very sparse BA graphs.

Moreover, newly from these plots across various parameters, we also observe that given same $n$, `Deg-Greedy` and `LTFT` tend to have lower percentage for sparse graphs (smaller $d$ or $m$) and higher percentage for denser graphs (larger $d$ or $m$) across all 3 parts. On the other hand, `OnlineMIS`, `ReduMIS`, `iSCO`, and `LwD` tend to have the percentage of 1st 1/3 decreases, while the percentage of the 2nd and 3rd 1/3 increases, when the density of graph increases ($d$ or $m$ increases for same $n$). This shows another qualitative difference between the algorithms similar to `Deg-Greedy` (`Deg-Greedy` and `LTFT`) and the good-performing algorithms (`OnlineMIS`, `ReduMIS`, `iSCO`, and `LwD`).

We also note that BA graphs $(n = 300, m = 50)$, $(n = 1000, m = 150)$, and $(n = 3000, m = 500)$ are outliers. Most algorithms have percentage close to 100% for all parts. This is because these graphs are rather different from other BA graphs. They have an easily found large MIS, which is the $m$ nodes initially in the graph at the start of the BA generation process. (Albert & Barabási, 2002). From Table 3, we can see many algorithms can find these MIS and report a MIS size of $m$ for these graphs. This suggests that for this special type of BA graphs, our serialization analysis can observe different characteristics from other BA graphs.

### C.3 Comparison between `Deg-Greedy` and `LwD`

`LwD`, similar to `LTFT`, is also a non-backtracking MDP based algorithm which picks nodes sequentially. The main difference is that instead of picking 1 node at a step like `Deg-Greedy` and `LTFT`, it picks some nodes at a step, which can be 0 or 1 or multiple nodes. In that case, it does not have a *natural serialization* like `LTFT` (discussed in Section 4.1). However, since it still have steps and we still know some nodes are chosen before others, we can perform a serialization within each step. We call this *pseudo-natural serialization.*

The procedure of our pseudo-natural serialization is as follows. Consider a step $t$ of `LwD`, let the independent set before the step be $\mathcal{I}_{t-1}$. Similar to Algorithm 1, we build a residual graph $\mathcal{G}'$ which removes the nodes in $\mathcal{I}_{t-1}$ and their neighbors from $\mathcal{G}$. Then, `LwD` chose a set of nodes $\mathcal{S}_t$ to add to the independent set. We then perform the serialization for the set $\mathcal{S}_t$ (replace $\mathcal{I}'$ by $\mathcal{S}$ in Algorithm 1) and get a ordered list $\mathcal{L}_t$. $\mathcal{L}_t$ is the ordered list for a step. We then concatenate all the ordered lists $\mathcal{L}_t$'s for all the steps in order to get a full ordered list $\mathcal{L}$. For this serialization, we do not repeat it as in Algorithm 1.

Similar to Section 4.1, we plot a heatmap Figure 6 across various parameters for ER and BA graphs on the average percentage of the nodes being the smallest degree node in the residual graph. In addition to that, similar to Section 4.2, we divide the ordered list $\mathcal{L}$ into 3 equal parts to compute the average percentage of the nodes being the smallest degree node seperately.

The heatmaps for overall percentages suggest that `LwD` is not very similar to `Deg-Greedy` like `LTFT`. From the heatmaps for different 1/3 parts, we can see the percentage increases from the 1st 1/3 to the 3rd 1/3 for all the datasets (different parameters of the synthetic graphs). This aligns with our "counterfactual" serialization results for `LwD` in Section 4.2, where we also observe the percentage increases clearly from 1st 1/3 to 3rd 1/3. This shows that our serialization method in Section 4.2 can reflect the pattern correctly.

The percentages here in the heatmap is smaller than the percentages for "counterfactual" serialization in the bar graphs in Section 4.2. This is likely due to the fact that in the "counterfactual" serialization we repeat the serialization process for 100 times and report the highest percentages we get, while here we only do serialization once for each step.

### C.4 Local search

Table 10 shows the full results after incorporating 2-improvement local search from the ARW local search algorithm (Andrade et al., 2012) as a post-processing step, which is discussed in Section 4.3.

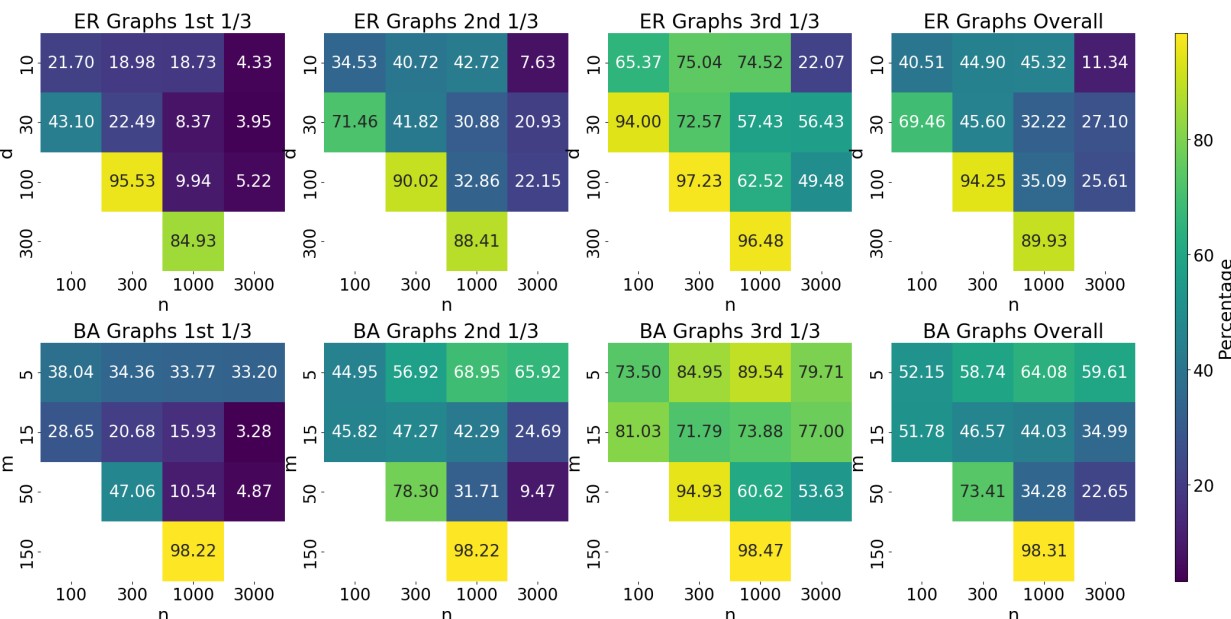

Figure 6: **Percentage of the smallest possible degree node in the pseudo-natural serialization of LwD, i.e., behaves similarly to degree-based greedy, for the 3 equal parts of the serialization, and the overal average** From the overall heatmaps, we can see LwD is not very similar to Deg-Greedy like LTFT. From the heatmaps for different 1/3 parts, we can see the percentage increases from the 1st 1/3 to the 3rd 1/3 for all the datasets (different parameters of the synthetic graphs). This aligns with our "counterfactual" serialization results for LwD in Section 4.2, where we also observe the percentage increases clearly from 1st 1/3 to 3rd 1/3.

Table 10: **Adding local search as a post-processing procedure.** This is the full graph for local search described in Section 4.3.

| param | Heuristics | | | GPU-acc | | Learning-based | | |
|---|---|---|---|---|---|---|---|---|
| | Deg-Greedy | OnlineMIS | ReduMIS | iSCO | PCQO | LwD | LTFT | DIFUSCO |
| ER Graphs | | | | | | | | |
| 100,10 | 29.50 (0.25) | 30.50 | 30.50 | 30.62 (0) | 30.63 (0) | 30.38 (0) | 29.00 (0.38) | 30.25 (0) |
| 100,30 | 13.62 (0) | 14.00 | 14.75 | 14.50 (0) | 14.50 (0) | 14.38 (0) | 13.25 (0.13) | 13.88 (0) |
| 300,10 | 92.12 (0.50) | 93.88 | 94.38 | 94.75 (0) | 94.63 (0) | 94.25 (0) | 90.12 (1.50) | 93.50 (0) |
| 300,30 | 44.75 (0.25) | 47.88 | 47.88 | 47.62 (0) | 47.63 (0) | 46.88 (0) | 44.00 (0.75) | 44.62 (0.74) |
| 300,100 | 16.12 (0) | 18.00 | 18.38 | 18.00 (0) | 18.00 (0) | 17.12 (0.12) | 16.25 (0) | 16.88 (0.26) |
| 1000,10 | 305.38 (2.13) | 314.75 | 316.13 | 315.62 (0) | 310.75 (0.75) | 311.25 (0) | 300.00 (3.00) | 306.38 (2.50) |
| 1000,30 | 152.75 (1.75) | 158.88 | 163.75 | 163.50 (0) | 159.13 (0.50) | 158.62 (0.24) | 151.38 (1.38) | 150.62 (6.87) |
| 1000,100 | 60.75 (0.13) | 64.75 | 66.63 | 66.50 (0) | 60.88 (0.75) | 64.12 (0.24) | 61.88 (1.00) | 57.88 (2.50) |
| 1000,300 | 22.25 (0) | 25.00 | 25.75 | 24.62 (0) | 23.25 (0) | 20.75 (1.63) | 22.62 (0) | 21.62 (0.74) |
| 3000,10 | 913.62 (6.50) | 947.25 | 954.25 | 950.88 (0) | 927.38 (4.13) | 935.38 (1.26) | 900.62 (12.37) | 918.00 (16) |
| 3000,30 | 456.12 (4.24) | 480.88 | 493.13 | 491.62 (0) | 469.75 (5.50) | 474.00 (0.75) | 454.38 (5.38) | 442.75 (29.37) |
| 3000,100 | 185.62 (2) | 194.38 | 201.50 | 200.38 (0) | 189.75 (4.12) | 191.50 (0.75) | 185.62 (1.62) | 182.25 (10.87) |
| 3000,300 | 74.00 (0.50) | 77.63 | 80.75 | 78.88 (0) | 70.88 (1.63) | – | 74.38 (0.50) | – |
| 3000,1000 | 23.38 (0) | 26.00 | 26.25 | – | 23.00 (0) | – | 23.62 (0) | – |
| BA Graphs | | | | | | | | |
| 100,5 | 39.25 (0) | 39.50 | 39.50 | 39.50 (0) | 39.50 (0) | 39.50 (0) | 38.50 (0.38) | 39.38 (0) |
| 100,15 | 21.00 (0) | 21.63 | 21.63 | 21.62 (0) | 21.63 (0) | 21.62 (0) | 20.62 (0) | 21.25 (0) |
| 300,5 | 122.38 (0.26) | 123.13 | 123.13 | 123.12 (0) | 123.00 (0) | 123.12 (0) | 118.62 (3.00) | 123.00 (0.12) |
| 300,15 | 70.00 (0.75) | 71.38 | 71.38 | 71.38 (0) | 71.25 (0) | 70.75 (0) | 66.62 (1.87) | 70.00 (0.50) |
| 300,50 | 39.25 (1.75) | 49.88 | 50.00 | 50.00 (0) | 50.00 (0) | 50.00 (0) | 43.62 (1.87) | 50.00 (0) |
| 1000,5 | 413.38 (2) | 417.13 | 417.13 | 417.12 (0) | 415.88 (0.13) | 416.00 (0) | 400.25 (14.50) | 417.12 (0) |
| 1000,15 | 234.88 (1.63) | 245.00 | 246.38 | 246.25 (0) | 242.12 (0.12) | 243.12 (0) | 230.50 (6.25) | 237.88 (1.50) |
| 1000,50 | 108.38 (1.00) | 115.75 | 116.88 | 116.75 (0) | 114.00 (0.50) | 113.12 (0.12) | 106.75 (0.37) | 108.75 (3.50) |
| 1000,150 | 90.88 (7.63) | 150.00 | 150.00 | 150.00 (0) | 150.00 (0) | 150.00 (0) | 87.62 (5.37) | – |
| 3000,5 | 1241.50 (4.75) | 1257.00 | 1257.13 | 1255.62 (0) | 1245.38 (2.38) | 1248.50 (0.38) | 1213.12 (35.87) | 1254.75 (0.37) |
| 3000,15 | 714.62 (7.50) | 749.63 | 754.50 | 752.00 (0) | 729.38 (7.13) | 730.50 (2.75) | 693.00 (31.12) | 731.75 (5.37) |
| 3000,50 | 339.00 (3.62) | 362.63 | 369.75 | 368.25 (0) | 359.25 (2.13) | 341.00 (4.12) | 334.00 (10.12) | – |
| 3000,150 | 142.88 (2.38) | 160.25 | 165.75 | 164.00 (0) | 155.38 (2.63) | – | 146.50 (1.75) | – |
| 3000,500 | 172.12 (9.12) | 500.00 | 500.00 | – | 491.00 (0) | – | 229.62 (5.74) | – |

### C.5 More results on the ratio

In addition to what we show in Section 4.4, Figure 7 shows the ratio of MIS size to $\frac{n \ln d}{d}$ for ER graphs with number of nodes $n$ and average degree $d$ across more algorithms. We can see that `ReduMIS`, `OnlineMIS`, and `iSCO` has consitently high ratios more than 1.2. `Ran-Greedy` stays around 1.0 for all $(n, d)$. Other algorithms, including `Deg-Greedy`, all have higher ratios for sparser graphs, but lower ratios (close to 1) for denser graphs.

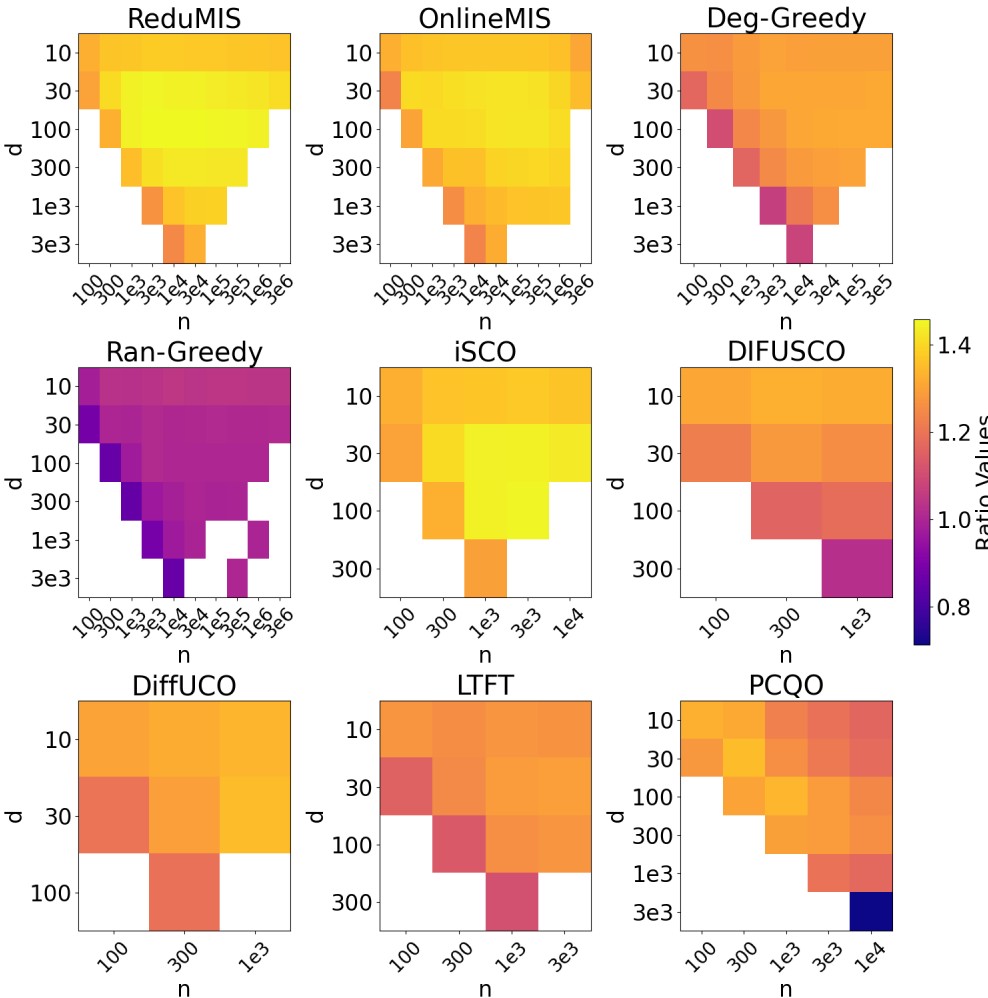

Figure 7: **Heatmap for ratios of MIS size to** $nln(d)/d$ **on ER graphs with number of nodes** $n$ **and average degree** $d$**.** We can see that `ReduMIS`, `OnlineMIS`, and `iSCO` has consitently high ratios more than 1.2. `Ran-Greedy` stays around 1.0 for all $(n, d)$. Other algorithms, including `Deg-Greedy`, all have higher ratios for sparser graphs, but lower ratios (close to 1) for denser graphs.

## D  Code and Data Release

We are committed to releasing the resources needed to reproduce our experiments. Our analysis code and random graph generation code is available at `https://github.com/yikai-wu/MIS-UnExp`.

This repository is a lightweight research-code release designed to support the full experimental pipeline of the paper rather than a standalone software package. It includes data generation, baseline solvers (greedy algorithms), analysis scripts, local search post-processing, and evaluation pipelines for learned methods. All

components operate on shared graph and solution formats (NetworkX `.gpickle` graphs and 0/1 solution vectors stored either as plain text or `.result` files) and are designed to be composed into a unified workflow.

## D.1 Data generation

The synthetic datasets in this work consist of large random graphs, including Erdős–Renyí (ER) graphs and Barabási–Albert (BA) graphs. Because many instances are extremely large, we do not release all generated graphs. Instead, we provide the exact generation code and parameters needed to reproduce them.

The `data/` folder extends the MIS benchmark framework (`https://github.com/MaxiBoether/mis-benchmark-framework`). Our code is designed to replace the corresponding files in the upstream `data_generation/` module while reusing the framework's infrastructure (sampling interface, dataset writing, and optional labeling). The script `random_graph.py` implements both samplers and dataset generation.

Supported graph families include Erdős–Renyí, `G(n,m)`, Barabási–Albert, Holme–Kim, Watts–Strogatz, and hyperbolic random graphs, as well as fixed-size variants such as `GND`, `Regular`, `BA_n_m`, `HK_n_m_p`, and `WS_n_k_p`. Graphs are serialized as `.gpickle` files and can optionally include node weights and labels.

For RB graphs, we use the generation code from the `LTFT` repository (`https://github.com/zdhNarsil/GFlowNet-CombOpt/tree/main/data`), following prior work.

We also evaluate on real-world datasets REDDIT-MULTI-5K and COLLAB from the TUDataset collection Morris et al. (2020) (`https://chrsmrrs.github.io/datasets/`).

## D.2 Greedy algorithms

The repository includes implementations of the baseline solvers used in the paper, namely degree-based greedy (`Deg-Greedy`) and random greedy (`Ran-Greedy`). These implementations operate directly on the shared data format (`.gpickle` graphs and 0/1 solution vectors) and are integrated with the overall pipeline.

Both `Deg-Greedy` and `Ran-Greedy` support repeated randomized runs with best-of-$k$ selection, matching the experimental protocol described in the paper. The local search implementation corresponds to the post-processing procedure used in Section 4.3, taking an initial solution and iteratively improving it while maintaining feasibility.

## D.3 Analysis codes

We release the analysis code used to aggregate raw solver outputs and regenerate the results reported in Section 4.

**LTFT comparison (`ltft/`).** The `ltft/` folder contains the evaluation entry point used for `LTFT`-based methods (see Section 4.1). The main script, `evaluate.py`, runs inference on MIS test graphs, performs repeated stochastic rollouts (`num_repeat=20`), and saves the best solution for each graph as a `.result` file. It also records degree-ranking diagnostics in `rankings.out` and `stats.out`. This script depends on the upstream implementation at `https://github.com/zdhNarsil/GFlowNet-CombOpt`.

**Other algorithm comparison (`serialization/`).** The `serialization/` folder contains the analysis scripts used for our serialization-style comparison to degree-based greedy (see Section 4.2). All scripts operate on `.gpickle` graphs and solution vectors (plain text or `.result`), validate independence, and simulate the randomized greedy-removal process. They report rank-1, top five percent, and top ten percent agreement statistics. Scripts include `compare_greedy.py`, `compare_greedy_segment.py`, and `compare_greedy_folders.py`.

**Local search analysis (`local_search/`).** The `local_search/` folder contains scripts used for analyzing the effect of local search (see Section 4.3), including evaluation of improvements over initial solutions and integration with the same solution format used across the pipeline.

### D.4 External solver codebases

In addition to our own implementations, we use external solvers from prior work:

- **KaMIS (OnlineMIS / ReduMIS)** Dahlum et al. (2016); Lamm et al. (2017): `https://github.com/KarlsruheMIS/KaMIS`.

- `LwD` Ahn et al. (2020): official repo `https://github.com/sungsoo-ahn/learning_what_to_defer`; we use the patched MISBenchmark version Böther et al. (2022) (`https://github.com/MaxiBoether/mis-benchmark-framework`).

- `LTFT` Zhang et al. (2023): `https://github.com/zdhNarsil/GFlowNet-CombOpt`.

- `DIFUSCO` Sun & Yang (2023): `https://github.com/Edward-Sun/DIFUSCO`.

- `DiffUCO` Sanokowski et al. (2024): `https://github.com/ml-jku/DIffUCO`.

- `PCQO` Alkhouri et al. (2025): `https://github.com/ledenmat/pCQO-mis-benchmark`.

- `iSCO` Sun et al. (2023): `https://github.com/google-research/discs`.

**Reproducibility statement.** The released repository includes data generation, baseline solvers (greedy algorithms), local search post-processing, analysis scripts, and evaluation pipelines, together with links to external solver implementations. These components are sufficient to reproduce the datasets, baseline results, and analysis presented in the paper. Documentation is provided in the repository to guide reproduction of the main experiments.

