# OpenReview forum: "Unrealized Expectations: Comparing AI Methods vs Classical Algorithms for Maximum Independent Set"
_TMLR — Accepted by TMLR_

### Review · Reviewer_3pmo · 2025-12-22

**Summary Of Contributions:**

This paper compares a bunch of AI algorithms for combinatorial optimization against traditional heuristics on the MIS (maximum independent set) problem, finding that AI algorithms do not seem to do much better than simple heuristics.

**Additional Comments:**

Note that I am not really an expert in combinatorial optimization / graph theory so many of my suggestions were things I learned while doing research for this review. I have a very superficial understanding of theory in this area.

**Audience:**

Yes

**Audience Explanation:**

Yes, ML for graph algorithms is an important topic which receives continued interest from the ML community.

**Broader Impact Concerns:**

No concerns

**Claims And Evidence:**

Yes

**Claims Explanation:**

Yes, as far as I can tell the key experiments were done correctly and the results support the claims. That being said, it would be highly beneficial to the community if the authors released all their evaluation code and raw data along with the paper.

~~The reason I answered "no" is just because I am concerned that the author's claim needs to be qualified with a statement about bias in the evaluation graphs (see "requested changes" below). If this is done I can change my answer to "yes".~~

**Update**: after discussion with the reviewers I am no longer concerned about graph bias, I've changed my answer to "yes".

**Requested Changes:**

Please clearly explain the limitations of your conclusions due to the choice of graphs. Although the paper touches on some of the theory behind ER random graphs, it really should explain the biases of these graph types:

- ER random graphs are highly connected, and with high probability have O(ln d) MIS (a rather small number). Furthermore, due to the law of large numbers there is a concentration in the degree of nodes, so from the perspective of a MIS solver the graph "looks" like a couple of low-degree nodes and a bunch of nodes with similar degree. Choosing the low degree nodes already gets close to the optimum.
- BA random graphs have "hubs" which are _not_ part of the MIS solution (because they are too highly-connected). Choosing non-hub node (which also have low degree) is a great strategy.
- The "real" graphs see roughly the same performance of most algorithms. I suspect their structure is closer to BA since many real-world networks have this kind of structure

In short, I think the set of evaluation graphs is biased towards graphs where greedy heuristics are rather effective. This isn't necessarily a bad thing if you care about performance on real-world graphs (maybe greedy heuristics tend to work very well on real world graphs), but since the paper's main claim is about the _performance of AI algorithms_ I think at a minimum this bias should be stated transparently. Ideally the authors could do an additional experiment on systematically "harder" graphs and see if their conclusion still holds.

I don't have a particular suggestion in mind for "harder" graphs, but some ideas for inspiration:

- Random geometric graphs / unit disk graphs (high degree of clustering)
- Stochastic block models / community graphs (tests whether heuristics are thrown off by community structures)
- Hard benchmark sets for maximum clique problems
- Random regular graphs (or graphs with a much more concentrated distribution of node degrees)


My second requested change is about §4.4. As far as I understand the results from Coja-Oghlan & Efthymiou which the authors claim to challenge are _asymptotic_, so experiments with small n and d don't contradict the theory. Maybe instead say something like "empirically, modest size graphs are very far from the asymptotic regime." The current phrasing seems to overclaim the significance of the finding.

---

> ### Author Response · Authors · 2026-02-08
>
> We thank the reviewer for the positive assessment of our experimental execution and the confirmation that our results support our claims. We appreciate the constructive feedback. We have addressed the specific concerns below.
>
> ### **1. Bias in Graph Choices and "Harder" Graphs**
>
> We do not think there is enough evidence to claim that ER and BA graphs are “biased” because “greedy heuristics are rather effective” on them. In fact, our results do not support the specific claim that “choosing the low-degree nodes already gets close to the optimum.” While Deg-Greedy is indeed a good baseline, it is far from optimal. **Table 1** and **Table 2** reveal that Deg-Greedy lags well behind the state-of-the-art *ReduMIS* for both ER and BA graphs. The reviewer is correct that AI methods (specifically *LTFT*) can learn to mimic degree-based heuristics, but our experiments show that they are nevertheless consistently outperformed by classical solvers such as *KaMIS*.
>
> Regarding the request for “harder” graphs, our evaluation already includes RB graphs (Model RB) [2]. As detailed in **Section 2.3**, RB graphs are widely considered “difficult instances … due to structured randomness and high solution hardness.” In **Table 3**, we show that *ReduMIS* consistently outperforms the best AI-inspired methods even on these intrinsically hard graphs. This demonstrates that the performance gap is robust and persists on graphs specifically designed to be resistant to simple heuristics.
>
> We thank the reviewer for suggesting other interesting families of randomly generated graphs (such as random geometric graphs or stochastic block models) and agree that these would be meaningful directions for future work. However, for this study, we prioritized the ER and BA families because they served as the standard benchmarks in the existing papers proposing the AI-based solvers we evaluate. By adhering to these established benchmarks, we demonstrate our primary point: previous works overstated the power of AI-based methods on their own chosen benchmarks.
>
> Furthermore, maintaining this focus allowed us to conduct detailed experiments that systematically vary graph size ($N$) and average degree ($d/m$). This analysis was crucial for revealing why AI methods degrade to simple greedy heuristics as graph density increases. We also note that *MISBench* [1] tested learning-based methods on a wider variety of graph types without this systematic variation but similarly found no decisive advantage for AI methods. Our work extends this by showing that the gap persists with the newest AI algorithms and is robust across scales.
>
> We added a new subsection, **Section 5.1**, to include this discussion.
>
> Finally, regarding real-world graphs, we indeed observe roughly similar performance across most algorithms. However, this is due to the fact that available real-world datasets suitable for training (i.e., those with sufficient quantity and size) are algorithmically “easy,” making it difficult to differentiate between sophisticated solvers and simple heuristics. We added the final paragraph in **Section 3** to include this discussion.
>
>
> ### **2. Asymptotic Theoretical Conjectures**
>
> We agree that the conjecture by Coja-Oghlan & Efthymiou [3] is asymptotic, so finite-graph experiments cannot refute it. Our original phrasing was that the conjecture “is either false or does not apply for real-life sizes,” which did not claim a contradiction or a refutation. However, to avoid misinterpretation, we revised **Section 5.2** (previously **Section 4.4**) to state: “One possible reasoning for our finding is that the graphs we are able to evaluate empirically are very far from the asymptotic regime where the conjecture might be applicable (i.e., if it is true).”
>
> ### **3. Code and Data Release**
>
> We are committed to releasing the resources needed for reproducibility. Specifically:
>
> - **Analysis:** We will release the analysis code used to generate the results and tables in this paper.
> - **Solvers:** We utilize codebases from previous works. We will add an Appendix listing the repositories for every method used.
> - **Data:** Since the generated synthetic graphs are massive, sharing raw files is logistically difficult. Instead, we will provide the exact generation scripts and parameters (utilizing the standard MISBench framework [1]) to allow exact reproduction of the graph distributions. We are happy to assist researchers in reproducing specific instances.
>
> We will update our manuscript to reflect these additions for code and data release.
>
> ### **References**
>
> [1] Böther, M., et al. *What’s wrong with deep learning in tree search for combinatorial optimization.* ICLR, 2022.
>
> [2] Xu, K. and Li, W. *Exact phase transitions in random constraint satisfaction problems.* JAIR, 12, 2000.
>
> [3] Coja-Oghlan, A. and Efthymiou, C. *On independent sets in random graphs.* Random Struct. Alg., 47(3), 2015.

---

> > ### Comment · Reviewer_3pmo · 2026-02-10
> > **Graph bias question**
> >
> > Thank you for the changes so far. Let's try to work out where we disagree during the discussion phase and come to an understanding.
> >
> > First: I am ok with the framing of re-using benchmarks from previous papers (basically evaluating the algorithms on their own terms). If this is the framing you'd like to use in your paper then I'm happy to recommend acceptance. However, in §1 you introduce the much broader question "Do AI-inspired algorithms perform better than classical heuristics for combinatorial optimization?" I you want to use this framing I think you also have the burden of arguing for the limitations/validity of the benchmarks, at least to some degree. I hope you also find this reasonable?
> >
> > Second: the framing of "state-of-the-art" in this context is odd. MIS is a number for a graph. There is a true correct answer, and either an algorithm gets it correct or not. However, this is not your evaluation: instead you output which algorithm produces the highest number. Higher numbers will of course be closer to the correct answer, but what is the context where "closer" answers are important? What is the marginal value of finding an MIS solution +1 higher than another method? Essentially I think the evaluation framework of "producing higher numbers" for a problem which has a single correct answer needs some justification.
> >
> > Third: you mention "significant" a lot. I assume you mean "statistically" significant, but I am talking about _practical_ significance. I'm not sure what is practical here. When I see Tables 1-3 what stands out most is that Deg-Greedy is always within 10% of the highest number. On smaller graphs it is often within 1 node of the best method. Without any context, that seems pretty good to me, I'm not sure how important this extra 10% of couple of nodes are. This context of a "practical significant difference" is required to make judgements about performance differences. This might be part of where we disagree. I'd appreciate you explaining your views here.
> >
> > Finally, on the issue of bias in the graphs. First, I don't believe it is possible to create an "unbiased" set of evaluation graphs, so my use of the word "bias" was not meant as a criticism. No need to be very defensive here, but you should be open about biases. When I wrote "rather effective" this was mostly noting that performance differences are small by my subjective judgement (without any problem context). I was suggesting other graph types in case there are graph classes where the performance difference between Deg-Greedy and other methods is 2x or 3x, instead of just ~10%.
> >
> > ### Next steps
> >
> > I'm happy to just finish the discussion in the comments rather than doing pdf revisions. I'd appreciate you responding to my points above, in particular:
> >
> > 1. Whether you want to narrow the frame of your question to be "better along previous axes of evaluation" rather than "better in general". If you want to keep the current framing, do you agree it's important to argue for the benchmark validity somewhat?
> > 2. What "state of the art" means for MIS and the context where suboptimal solutions are judged (and whether a ~10% difference is meaningful)
> >
> > I'll prioritize responding to your answers within 1-2 days.

---

> > > ### Author Response · Authors · 2026-02-16
> > >
> > > We thank the reviewer for the thoughtful follow-up. We acknowledge potential misunderstandings regarding the specific evaluation context of this field and clarify our position below.
> > >
> > > We also note that we have removed the previously added Section 5.1 on potential bias in graph choices, as the reviewer’s follow-up comments suggest a different point of concern than what that section was addressing. We are happy to revise the manuscript further if needed once we reach agreement on the appropriate framing.
> > >
> > > ### **1. "State of the Art" (SOTA) and the "Producing Higher Numbers" Metric**
> > >
> > > "Producing higher numbers" (finding a larger independent set) is the standard performance metric in combinatorial optimization for NP-hard problems where exact solutions are intractable. For graph sizes considered in modern benchmarks, the true optimum is typically unknown and computationally infeasible to obtain. In this setting, relative comparison of feasible solution sizes is the accepted and established evaluation framework across both classical [2] and AI communities [1, 3, 4]. Given this broad consensus in the literature, this evaluation framework reflects established standard practice in MIS research rather than a design choice specific to our study.
> > >
> > > Accordingly, "State of the Art" is defined by the solver that consistently finds the largest valid solutions on broad benchmarks under comparable computational budgets. *KaMIS* (specifically *ReduMIS*) is widely regarded as the SOTA heuristic because it consistently finds the largest independent sets [1].
> > >
> > > ### **2. Practical Significance of a ~10% Gap**
> > >
> > > In MIS literature, a 10% improvement is widely accepted as significant [1, 2]. As algorithms approach theoretical limits, gains become difficult to obtain; bridging the gap between a naive heuristic and SOTA often requires sophisticated engineering rather than simple changes. Notably, recent AI-based MIS papers—including diffusion-based methods such as DiFusCO and DiffUCO—also treat single-digit percentage improvements as meaningful advances over prior AI baselines [3, 4, 5, 6], and many reported gains are well below 10%. Within this established context, a gap on the order of ~10% relative to SOTA is therefore substantial rather than marginal.
> > >
> > > Practical judgment depends on the specific use case, and this study does not aim to prescribe a universal rule. For instance, if a user strictly prioritizes speed, *Deg-Greedy* is likely best for the graph types we evaluated. However, if solution quality ("the number") matters, *KaMIS* is the best choice. Existing AI methods currently fail to dominate either end of this spectrum—they are computationally expensive (requiring GPUs) without matching the performance of *KaMIS*. There are nevertheless possible niche use cases where current AI-based methods *may have an edge*, although not revealed in our study.
> > >
> > > ### **3. Graph Bias and Scope of "2x or 3x" Differences**
> > >
> > > Regarding the suggestion to search for graph classes with "2x or 3x" differences, our evaluation already includes regimes where simple heuristics degrade substantially. For example, on the densest BA graphs, the performance gap between *Deg-Greedy* and *ReduMIS* exceeds 10%, demonstrating that we do not restrict attention to instances where greedy is trivially near-optimal. Yet even in these regimes, AI-based methods do not surpass SOTA solvers.
> > >
> > > We also evaluate RB Graphs, which are widely regarded as structurally hard instances where simple heuristics such as *Deg-Greedy* typically do not perform well [4]. Even on these hard instances, the gap between *Deg-Greedy* and *ReduMIS* stays within roughly 10%. This suggests that a ~10% difference is already large in this problem class: even moving from "easy" to "hard" benchmark families does not typically produce gaps much larger than this. AI-based methods still do not outperform SOTA. This indicates that the observed gap is not an artifact of choosing instances that favor simple degree-based heuristics.
> > >
> > > While all benchmark choices introduce structural biases, our empirical results do not support the hypothesis that classical solvers outperform AI methods merely because the benchmarks favor greedy strategies. Finally, MISBench [1] evaluated learning-based methods across a broader variety of graph families and similarly found no decisive advantage for AI-based approaches.

---

> > > ### Author Response · Authors · 2026-02-16
> > >
> > > ### **4. Framing: "Better in General" vs. "Previous Axes of Evaluation"**
> > >
> > > "Do AI-inspired algorithms perform better than classical heuristics…?" is indeed a broad research question, as noted in our Introduction. However, empirical research requires clearly defined evaluation settings. Our graph families (ER, BA, RB) are not arbitrary choices but largely the union of benchmark families used in prior AI-based MIS papers [1, 3, 4]. We therefore evaluate AI methods along axes of evaluation that are widely accepted by the community and frequently used to support performance claims in the literature.
> > >
> > > "Better in general" is not a precisely defined notion without specifying distributions, scales, or computational constraints. If it refers to performance on these community-accepted benchmarks, our results indicate that classical heuristics remain superior. If it refers to practical utility, the answer necessarily depends on the application context and resource constraints. For example, *Deg-Greedy* prioritizes efficiency, whereas *KaMIS* prioritizes solution quality. Our study does not claim superiority in all possible settings; rather, it evaluates AI-based methods under the same benchmark axes on which their performance has traditionally been reported.
> > >
> > > ### **References**
> > >
> > > [1] Böther, M., et al. "What's wrong with deep learning in tree search for combinatorial optimization." *ICLR*, 2022.
> > >
> > > [2] Lamm, S., et al. "Finding near-optimal independent sets at scale." *Journal of Heuristics*, 2017.
> > >
> > > [3] Ahn, S., et al. "Learning what to defer for maximum independent sets." *ICML*, 2020.
> > >
> > > [4] Zhang, D., et al. "Let the flows tell: Solving graph combinatorial problems with GFlowNets." *NeurIPS*, 2023.
> > >
> > > [5] Sun, Y., et al. "DiFusCO: Diffusion Solvers for Combinatorial Optimization." *NeurIPS*, 2023.
> > >
> > > [6] Sanokowski, S., et al. "DiffUCO: Diffusion for Unsupervised Combinatorial Optimization." *ICLR*, 2024.

---

> > > > ### Comment · Reviewer_3pmo · 2026-02-20
> > > > **Thanks for the response, I will recommend acceptance**
> > > >
> > > > Thanks for the quick follow up to my questions and sorry it took more than 2 days for me to respond.
> > > >
> > > > I see your broader point: for better or worse, this is the standard evaluation framework in this subfield, and you are mostly following the conventions and norms for performance difference. You show that within this framework AI-based methods don't work as well as expected. I am happy with this contribution, especially since the primary readers for this paper will probably be from the same subfield. Therefore I've changed my answer to "supporting claims" from "no" to "yes" and will recommend acceptance.
> > > >
> > > > That being said, even though I concede that it's not your responsibility to justify and explain the norms / evaluation practices of this subfield, I encourage you to think more deeply about this. "Accepted norms" is not itself a strong argument. In my own subfield (part of AI for science), I witnessed very contrived benchmarks become entrenched and "gamed" until they were completely irrelevant to any practical application. The original benchmark creator had good intentions but simply chose the most convenient quantitative evaluation strategy for their paper. Subsequent papers just cited the original paper, and before too long a very silly benchmark was the "accepted norm." Despite the apparent "consensus" in the field, practitioners / domain experts didn't take the results seriously. Eventually interest moved elsewhere, but overall this probably hurt the field.
> > > >
> > > > Of course, since this is a blind review process I don't know the career stage of the authors. Nonetheless, let me try to give career advice. When applying for jobs (both industry and academia), your work will be read and questioned by people outside of your subfield who don't share the same set of "accepted norms". They might challenge these norms- especially in industry where evaluations are geared more heavily to a specific application and may differ significantly from what is done in academia. You will look naive if your answer is just "I don't know, these are the accepted norms". In my own career, my answer has been "yes I know the standard evaluation is bad, I tried to do work whose core value proposition was not just empirical performance". I encourage you to have a thoughtful answer prepared for these types of questions in the future!

---

### Review · Reviewer_e7Pk · 2026-01-03

**Summary Of Contributions:**

AI-inspired approaches for combinatorial optimization are often consistently outperformed by state-of-the-art classical solvers. The authors investigate the reasons behind this gap, proposing a novel analysis framework of analysis called serialization. Their analysis shows that non-backtracking AI-inspired methods effectively perform reasoning similar to simple degree-based greedy heuristics, which explains their inferior performance compared to advanced solvers such as KaMIS..

**Additional Comments:**

This paper makes a clear and well-supported contribution. The results are informative and the insights might influence future research lines.

**Audience:**

Yes

**Audience Explanation:**

The paper can be interesting for the TMLR audience as it introduces informative results whose methods may influence future lines of research outside the scope of the paper itself

**Broader Impact Concerns:**

No broader immpact concern

**Claims And Evidence:**

Yes

**Claims Explanation:**

The paper seems sound, the scope is clear and the evidence appears convincing

**Requested Changes:**

The TMLR guidelines suggest to be cautios about overstatements about the results of the paper, it appears to me that, although the paper has many merits, it seems to me too speculative to assert that KaMIS’s performance “contradicts” or challenges the Coja-Oghlan Efthymiou conjecture, can the authors explain more thoriughly this claim?

The authors note that PCQO is sensitive to hyperparameters and that better tuning might improve results. This should be discussed more critically: if AI methods require extensive tuning to match classical heuristics, that is itself a limitation, can be this due to the fact  that methods are under-optimized relative to KaMIS?

Serialization is intuitive I wonder if a more deepened theoretical analysis is possible.

---

> ### Author Response · Authors · 2026-02-08
>
> We thank the reviewer for the encouraging feedback and for recognizing the clarity and contribution of our work. We are glad that you found the “serialization” analysis framework novel and informative. We have addressed the specific suggestions below.
>
> ### **1. Asymptotic Theoretical Conjectures**
>
> We want to clarify that the original manuscript states that the conjecture “is either false or does not apply for real-life sizes,” which did not claim a contradiction or a refutation. Our intent was to highlight the discrepancy between theoretical bounds and empirical performance on finite graphs. To avoid misinterpretation, we revised **Section 5.2** (previously **Section 4.4**) to explicitly clarify this distinction, stating: “One possible reasoning for our finding is that the graphs we are able to evaluate empirically are very far from the asymptotic regime where the conjecture might be applicable (i.e., if it is true).”
>
> ### **2. Hyperparameter Sensitivity and Optimization**
>
> We agree that the sensitivity of AI methods such as *PCQO* to hyperparameters is a significant limitation and represents another weakness compared to classical heuristic methods. We also agree that AI implementations may be less optimized than *KaMIS*. However, since we either conducted hyperparameter searches or utilized the authors’ recommended settings (which are based on their own searches), the persistent and large performance gap suggests that the issue is more likely algorithmic rather than due to insufficient tuning.
>
> ### **3. Theoretical Analysis of Serialization**
>
> We appreciate the interest in the theoretical underpinnings of serialization. We agree that developing a rigorous theoretical analysis of these patterns is a valuable and promising direction for future work.

---

### Review · Reviewer_FE6x · 2026-02-02

**Summary Of Contributions:**

The present paper provides a large-scale, systematic benchmark showing that state-of-the-art AI-inspired methods for Maximum Independent Set (MIS) consistently underperform classical CPU-based solvers, especially KaMIS, even on in-distribution random graphs. In particular, they show that performance gaps widen with graph size and density, and that several AI methods fail to beat even the simplest degree-based greedy heuristic.

In order to analyze the performance, they introduce serialization, a new analysis tool that converts any MIS solution into a sequential decision process, enabling fine-grained comparison across algorithms. This analysis shows that some learning-based methods effectively replicate degree-based greedy behavior, explaining their limited gains.

On another angle, they present empirical evidence against the practical relevance of the shattering-threshold conjecture for MIS.

**Additional Comments:**

As you already noticed, AI-based methods perform especially well in real-world datasets. You mention this is because the datasets are easier. This is an important practical point: if the performance in real-world tasks is similar, what is the disadvantage of AI-based methods? Running time, computational resources? Are there no harder real-world datasets to compare? This point requires further discussion.

Regarding references and discussion, it seems like the paper extends the analysis of [1] to more datasets and deep learning-based algorithms. I reckon the paper is cited, but given its importance for this study, further discussion regarding its limitations is needed.
Furthermore, one of the conclusions of the paper is the need for "Principled Integration of Classical Heuristics" into deep learning frameworks. Maybe some discussion of current literature on the topic (such as [2]) could be helpful.


[1]- Angelini, M. C., & Ricci-Tersenghi, F. (2023). Modern graph neural networks do worse than classical greedy algorithms in solving combinatorial optimization problems like maximum independent set. Nature Machine Intelligence, 5(1), 29-31.

[2]- Garmendia, A. I., Ceberio, J., & Mendiburu, A. (2023). Neural improvement heuristics for graph combinatorial optimization problems. IEEE Transactions on Neural Networks and Learning Systems.

**Audience:**

Yes

**Audience Explanation:**

This would interest most researchers working on the intersection between machine learning and combinatorial optimization.

**Broader Impact Concerns:**

Not applicable.

**Claims And Evidence:**

Yes

**Claims Explanation:**

The paper's claims are mainly empirical, and the experiments are systematic: strong benchmarks, large and diverse datasets, and a wide range of methods are compared.

**Requested Changes:**

Small typo: In legend for Table 5, Incoperating->Incorporating

---

> ### Author Response · Authors · 2026-02-08
>
> We thank the reviewer for the positive assessment of our benchmark’s scale and systematic approach, and for recognizing the value of the serialization analysis. We appreciate the insightful comments regarding real-world datasets and related work. We have addressed the specific requests below.
>
> ### **1. Performance on Real-World Datasets**
>
> We appreciate this important practical question: if performance is similar on real-world datasets, what is the disadvantage of AI methods?
>
> The main disadvantages are running time and computational resources. Even the simplest degree-based greedy algorithm (*Deg-Greedy*) achieves optimal or near-optimal results on these graphs. *Deg-Greedy* runs on a single CPU thread and is orders of magnitude faster than AI-based methods, which typically require expensive GPUs for both training and inference. Even the state-of-the-art heuristic *KaMIS*, which excels on harder instances, is not as efficient as *Deg-Greedy* on these easy datasets. Therefore, for these specific real-world tasks, *Deg-Greedy* is the optimal choice.
>
> Regarding “harder” real-world datasets, we faced a scarcity of datasets that are both suitable for training (i.e., of sufficient quantity and size) and intrinsically hard. If the “easy” datasets we identified are truly representative of real-world tasks, this would imply that sophisticated algorithms—whether AI-based or advanced classical solvers—are largely unnecessary in practice. As the community generally relies on random graphs (ER and BA) as proxies for real-world tasks, we followed this standard and used them as our main evaluation graphs.
>
> We added the final paragraph in **Section 3** to discuss this point explicitly.
>
> ### **2. Related Work: Angelini & Ricci-Tersenghi [1]**
>
> Regarding Angelini & Ricci-Tersenghi [1], we would like to clarify that our work is not a direct extension of their specific evaluation. Their paper was a commentary evaluating a single AI-based algorithm (“Combinatorial Optimization with Physics-Inspired Graph Neural Networks” [2]) and found that it did not outperform the simplest degree-based greedy heuristic. Although their title (“Modern graph neural networks do worse…”) appears to suggest a broad conclusion about GNN-based methods, their empirical evaluation was limited to that specific algorithm.
>
> This limitation is therefore clear: it evaluates only one particular AI-based approach. We have updated **Section 1** and **Section 6** (previously **Section 5**) to clearly state this distinction.
>
> In addition, we consider Böther et al. [3] to be the most relevant prior benchmark study, as it focused on tree search, whereas our work covers a wider array of recent learning-based and direct optimization methods.
>
> ### **3. Related Work: Garmendia et al.**
>
> Regarding Garmendia et al. [4], we agree that this paper is highly relevant to our proposal. We have included it in **Section 7** (previously **Section 6**).
>
> ### **4. Typo**
>
> We thank the reviewer for pointing this out, and we will correct the typo.
>
> ### **References**
>
> [1] Angelini, M. C., & Ricci-Tersenghi, F. *Modern graph neural networks do worse than classical greedy algorithms in solving combinatorial optimization problems like maximum independent set.* Nature Machine Intelligence, 5(1), 29–31, 2023.
>
> [2] Schuetz, M. J., Brubaker, J. K., & Katzgraber, H. G. *Combinatorial optimization with physics-inspired graph neural networks.* Nature Machine Intelligence, 4(4), 367–377, 2022.
>
> [3] Böther, M., Kißig, O., Taraz, M., Cohen, S., Seidel, K., and Friedrich, T. *What’s wrong with deep learning in tree search for combinatorial optimization.* ICLR, 2022.
>
> [4] Garmendia, A. I., Ceberio, J., & Mendiburu, A. *Neural improvement heuristics for graph combinatorial optimization problems.* IEEE Transactions on Neural Networks and Learning Systems, 2023.

---

### Author Response · Authors · 2026-03-13

Dear Action Editor and Reviewers,

Thank you very much for your careful reading of our manuscript and for the thoughtful and constructive feedback throughout the review process. We sincerely appreciate the time and effort that the Action Editor and all reviewers devoted to evaluating our work.

We are grateful for the positive assessment and for the opportunity to publish our work in TMLR. We will proceed with preparing the camera-ready version shortly.

All suggested modifications discussed during the review process have already been incorporated into the current revision, with the exception of the public release of code and data. We are in the process of finalizing these materials and will include the corresponding links and documentation in the camera-ready version.

Thank you again for your support and for the helpful feedback that has improved the quality and clarity of the paper.

---

### Author Response · Authors · 2026-04-01

Dear Action Editor and Reviewers,

Thank you again for your thoughtful feedback and support throughout the review process.

We have now submitted the camera-ready version of our manuscript. In this version, we have included the finalized code and data release, along with an updated and detailed description in Appendix D.

We would greatly appreciate it if you could let us know whether any further edits or clarifications are needed at this stage.

Thank you once again for your time and for the valuable feedback that has helped improve our work.

---

### Decision · Action_Editor_AnSv · 2026-03-02

**Recommendation:** Accept as is

**Additional Comments:**

The authors addressed the concerns raised by all Reviewers and the Action Editor in the revised draft. Reviewer 3pmo initially raised various concerns regarding the evaluation and the benchmarks, but the authors explained in detail that they have followed the standard practices in their field, and convincingly argued that the difference between AI-inspired and classical methods is in fact significant.

Furthermore, the massive computational resources required by AI-inspired methods are a significant shortcoming, even in the case where they perform comparably to the classical solvers.

**Audience:**

Yes

**Audience Explanation:**

I believe the findings of this paper would be of interest to a broader audience. The paper is able to show through a very extensive experimentation that for the classical NP-hard problem of the Maximum Independent Set, traditional state-of-the-art solvers running on a single CPU consistently outperform AI-inspired methods, despite the fact that the latter often come with a much higher computational cost and can only handle relatively smaller graphs. Given the rapid advances in the field of AI, this research points to the need of well-designed evaluation benchmarks that show not only the strengths but also the weaknesses of modern AI-inspired frameworks compared to classical solvers. Such critical thinking is very important for the ML/AI community at a time when AI systems receive so much positive attention about their strengths.

**Claims And Evidence:**

Yes

**Claims Explanation:**

The paper deals with a very interesting question: are modern AI-inspired methods able to outperform classical solvers for combinatorial optimisation problems? The authors choose one of the most well-studied NP-hard problem as a representative, namely, the Maximum Independent Set problem. They design a very extensive experimental evaluation using different types of random graphs but also real-world graphs, with different graph sizes and degrees for the various graphs. The experiments consistently show that machine-learning based methods generally lag behind classical methods; GPU-accelerated methods, on the other hand, can sometimes achieve close to optimal performance but at a much higher computational cost, which limits their scalability. Overall, the state-of-the-art classical solves almost always perform the best at a fraction of the cost, and even the simple greedy algorithm can often outperform much more complex and resource-intensive ML-based methods.

To elucidate the source of these failures, the authors introduce a novel technique, serialization, which reveals that non-backtracking AI-inspired methods end up reasoning similarly to the simplest degree-based greedy, and thus worse than the state-of-the-art classical solvers. The authors also made a connection to the Coja-Oghlan & Efthymiou (2015) conjecture. All Reviewers agreed that the connection was weak and rather unsubstantiated due to the asymptotic nature of the conjecture, but the authors reworded the statement in their revision.

One of the concerns raised by the Reviews was that performance on real-world datasets was close for all methods. The authors explained that this is because "most real-world datasets suitable for training (i.e., those with sufficient quantity and size) are algorithmically “easy,” making it difficult to differentiate between sophisticated solvers and simple heuristics:". They also clarify that the have followed standard evaluation practices, and their benchmarks can capture not only simple but also structurally hard instances.